# Language Model-Enhanced Message Passing for Heterophilic Graph Learning

## Abstract

Traditional graph neural networks (GNNs), which rely on homophily-driven message passing, struggle with heterophilic graphs where connected nodes exhibit dissimilar features and different labels. While existing methods address heterophily through graph structure refinement or adaptation of neighbor aggregation, they often overlook the semantic potential of node text, rely on suboptimal message representation for propagation and compromise performance on homophilic graphs. To address these limitations, we propose a novel language model (LM)-enhanced message passing approach for heterophilic graph leaning (LEMP4HG). Specifically, for text-attributed graphs, we employ LM to generate connection analysis with paired node texts, which are encoded and then fused with paired node textual embeddings through a gating mechanism. The synthesized messages are semantically enriched and adaptively balanced with both nodes' information, which mitigates contradictory signals when neighbor aggregation in heterophilic regions. Furthermore, we introduce an active learning strategy guided by our heuristic MVRD (Modulated Variation of Reliable Distance), selectively enhancing node pairs suffer most from message passing, reducing the cost of analysis generation and side effects on homophilic regions. Extensive experiments validate that our approach excels on heterophilic graphs and performs robustly on homophilic ones, with a simple graph convolutional network (GCN) backbone and practical budget.

## 1 Introduction

Graph-structured data, which represents entities and their relationships through nodes and edges, are ubiquitous across diverse real-world domains (Frasconi et al., 1998; Goller & Kuchler, 1996). To enhance graph-based task performance, various GNNs have been developed, with traditional models relying on message passing mechanisms that update node representations by aggregating neighbor features, implicitly assuming homophily (McPherson et al., 2001), where connected nodes tend to share similar attributes and identical labels (Pei et al., 2020; Hamilton, 2020). However, these methods fail on heterophilic graphs (Pei et al., 2020; Zhu et al., 2020; Bo et al., 2021; Luan et al., 2022; 2021), where connected nodes often exhibit dissimilar features and different labels (Lozares et al., 2014). The issue lies in the indiscriminate neighbor aggregation, which introduce noisy or contradictory signals, compromising the quality of learned representations. (Zhu et al., 2020; 2021)

Existing efforts to address heterophily in GNNs can be broadly classified into the graph structure refinement and GNN architecture adaptation, respectively. The former refines the node's receptive filed by including non-local, multi-order, and potentially connected neighbors, or excluding unbefitting ones. For example, SEGSL (Zou et al., 2023) refines graph topology using structural entropy and encoding trees, while DHGR (Bi et al., 2024) add homophilic edges and prune heterophilic ones based on label or feature distribution similarity. GNN-SATA (Yang et al., 2024) introduce a soft association between topology and attributes to dynamically remove or add edges. The latter adapts the message passing and representation updating functions for heterophilic situation. For example, OGNN (Song et al., 2023) update node representation with multi-hop neighbors by orders, while EGLD (Zhang & Li, 2024) utilize dimension masking to balance the contributions of low and high-pass filtered features. LLM4HeG (Wu et al., 2024) employ LM to encode node texts and leverage its semantic understanding to discriminate edges to guide reweighting. EG-GCN (Liu et al., 2025) co-trains a edge discriminator with group graph convolution applied to divided neighborhoods.

However, these methods still have limitations: (1) Most ignore the semantics of node text, encoding node features with shallow embedding method like bag-of-words and FastText (Grave et al., 2018). Only LLM4HeG (Wu et al., 2024) leverage LM to unlock the deeper insight under heterophily semantically, but remains underexplored. (2) Their GNN architectures still rely on ineffective message representations derived from source node features, inevitably leading to signal conflicts in the heterophilic regions. (3) Some sacrifice performance on homophilic graphs to achieve success on heterophilic ones (Zhu et al., 2020; Chien et al., 2020; Lim et al., 2021; He et al., 2021a).

In this work, we delve into further integration between LM and GNN, and rethink the underlying message passing mechanism for heterophilic text-attributed graphs with small language model (SLM)-encoded node embeddings. Specifically, we aim to address the following research questions.

**RQ1:** *can LM effectively generate messages for passing between connected nodes?* Leveraging the prior knowledge and semantic understanding of LM, existing work has achieved great success on TAG tasks (Fan et al., 2024; Ren et al., 2024). Fundamentally, we utilize a SLM to encode all textual content. Given the paired node texts, LM can capture their key points, similarities, and distinctions, providing the connection analysis, which are encoded as preliminary messages. However, the static nature of these preliminary messages may hinder long-range neighbor aggregation and cause misalignment with node textual embeddings. Thus, we propose a gating mechanism to fuse preliminary messages with source and target node textual embeddings for propagation. The synthesized messages are semantically enriched and adaptively balanced with source and target nodes' information, which mitigates contradictory signals when neighbor aggregation in heterophilic regions.

**RQ2:** *how to avoid impractical cost of full-scale message enhancement by LM?* Enhancing message representations for all edges by LM incurs $O(E)$ complexity, where $E$ is the number of edges, making deployment costly. Thus, we adopt active learning (Cai et al., 2017), which improves performance by selectively querying labels for the most informative samples. Inspired by advancements like LLM-GNN (Chen et al., 2023), we adaptively propose querying LM for connection analysis of node pairs selected by our designed heuristic MVRD, which captures representation distortion from message passing. This significantly reduces the overhead of analysis generation and mitigates side effects on homophilic graphs by focusing on node pairs suffer most from message passing.

**RQ3:** *how to fairly evaluate a heterophily-specific model?* A good heterophily-specific graph model should excel on heterophilic graphs while maintaining at least parity on homophilic ones—often neglected in prior work. Existing homophily metrics (Abu-El-Haija et al., 2019; Pei et al., 2020; Luan et al., 2023) fail to reliably identify challenging graph datasets, where the pattern is more complex than "homophily wins, heterophily loses" (Luan et al., 2022; 2023). As Luan et al.(Luan et al., 2024), we evaluate 16 homogeneous TAG datasets with SLM-encoded node features, spanning diverse domains and homophily level. Each dataset is assessed with paired graph-aware and graph-agnostic models (e.g. 2-layer MLP&GCN), and categorized based on performance shift induced by message passing. We additionally evaluate baselines and ours from such an perspective.

In summary, our main contributions are as follows:

- We propose LEMP4HG, a novel LM-enhanced message passing approach for heterophilic graph, which encode and fuse LM-generated textual content with paired node texts to obtain enhanced message representations for propagation between connected nodes.

- We propose an active learning strategy guided by our heuristic MVRD, selectively enhance paired nodes suffering most from message passing measured by embedding shift, which reduces the cost of textual content generation and side effects on homophilic regions.

- Extensive experiments on 16 real-world datasets demonstrate that our LEMP4HG excels on heterophilic graphs and also delivers robust performance on homophilic graphs.

## 2 RELATED WORK

**Graph Neural Networks for Heterophily** Existing GNNs for heterophilic graphs mainly adopt two strategies: graph structure refinement and GNN architecture refinement. The former optimizes node receptive fields by selectively expanding neighborhood and pruning unbefitting connections. For example, Geom-GCN (Pei et al., 2020) extends neighborhood by embedding-based proximity, while U-GCN (Jin et al., 2021) extract information from 1-hop, 2-hop and k-nearest neighbors simul-

taneously. SEGSL (Zou et al., 2023) refines graph topology using structural entropy and encoding trees, while DHGR (Bi et al., 2024) based on label or feature distribution similarity. GNN-SATA (Yang et al., 2024) dynamically adds or removes edges by associating topology with attributes. The latter optimizes the neighboring aggregation and representation updating functions. For example, FAGCN (Bo et al., 2021) discriminatively aggregate low-frequency and high-frequency signals, while OGNN (Song et al., 2023) updates node representation with multi-hop neighbors by orders. EGLD (Zhang & Li, 2024) utilize dimension masking to balance low and high-pass filtered features. However, these methods rely on shallow text encodings, e.g. bag-of-words, overlooking semantics. In contrast, we encode text using SLM and use another LM to fully exploit semantic information.

**Language Model for Graph Learning**   Existing works integrating LM into graph tasks achieve great success (Fan et al., 2024; Ren et al., 2024), with three main methods. (Li et al., 2023) (1) LM as enhancer (Tan et al., 2023; Liu et al., 2023), where LM generate text and embeddings to enhance GNNs classifier. For example, TAPE (He et al., 2023) improves node representations with SLM-encoded text embeddings and LM-generated explanation for classification and pseudo labels. (2) LM as predictor, where graph structures are transformed into textual descriptions (Zhao et al., 2023; Ye et al., 2023), or textual features are combined with GNN-encoded structural information (Chai et al., 2023; Tang et al., 2024) for LM inference. (3) GNN-LM alignment, which aligns GNN and LM embeddings through contrastive learning (Radford et al., 2021), interactive supervision (Zhao et al., 2022), or GNN-guided LM training (Mavromatis et al., 2023). However, these methods are not tailored for heterophily. Only LLM4HeG (Wu et al., 2024) uses LM for edge discrimination and reweighting, yet the integration of LM for heterophily remains underexplored. In this work, we investigate using LM-generated texts to enhance message representation for propagation in GNNs.

**Graph Active Learning**   Traditional graph active learning (Cai et al., 2017) selects nodes and query labels to improve test performance within a limited budget. Existing researches mainly optimize selection strategy from multiple perspectives, such as the diversity and representativeness of the selected nodes (Zhang et al., 2021b). For example, Ma et al.(Ma et al., 2022) select nodes from distinct communities for broad coverage, while Zhang et al.(Zhang et al., 2021a) prioritize the nodes with higher influence scores. Some approaches employ reinforcement learning to optimize model accuracy (Hu et al., 2020a; Zhang et al., 2022). With the prevalence of LM, LLM-GNN (Chen et al., 2023) enhances node selection by using LM as annotators, addressing limited ground truth and noise in annotations. However, these works are limited to node label annotation, whereas message "annotation" between node pairs is equally crucial for better pattern learned by graph models.

## 3 METHODOLOGY

### 3.1 PRELIMINARY

**Problem Definition**   We define a graph $\mathcal{G} = (\mathcal{V}, \mathcal{E})$, where $\mathcal{V} = \{v_1, v_2, \ldots, v_N\}$ is the set of nodes, and $\mathcal{E} = \{e_{ij} \mid i \neq j\}$ is the set of edges without self-loops. The adjacency matrix $\mathcal{A} = (A_{i,j}) \in \mathbb{R}^{R \times R}$. $A_{i,j} = 1$ if there is an edge between nodes $v_i$ and $v_j$, otherwise $A_{i,j} = 0$. For text-attributed graph $\mathcal{G}^T = (\mathcal{V}, \mathcal{E}, \mathcal{T})$, $\mathcal{T} = \{t_1, t_2, \ldots, t_N\}$ denotes the textual contents and $\mathcal{X} = \{\boldsymbol{x_1}, \boldsymbol{x_2}, \ldots, \boldsymbol{x_N}\}$ denotes the SLM-encoded textual embeddings of $\mathcal{T}$. Each node $v_i \in \mathcal{V}$ is associated with one peice of text $t_i$ and its corresponding embedding $\boldsymbol{x_i}$. In this paper, we focus on the task of transductive node classification on text-attributed graphs in a semi-supervised way.

**Classic Message Passing Mechanism**   The classic message passing mechanism in GNNs involves two key steps: the neighboring aggregation and the update of node representations. For a node $v_i$, the process in $l$-layer is formalized as:

$$\boldsymbol{m}_i^l = \sigma(AGGR(\{\boldsymbol{h}_j^{l-1} \mid v_j \in \mathcal{N}(v_i)\})), \ \boldsymbol{h}_i^l = UPDATE(\boldsymbol{h}_i^{l-1}, \boldsymbol{m}_i^l) \tag{1}$$

where $\boldsymbol{h}_j^{l-1}$ is the representation of neighbor $v_j$, $\mathcal{N}(\cdot)$ is neighborhood function, $\boldsymbol{m}_i^l$ is aggregated message for node $v_i$ and $\boldsymbol{h}_i^l$ is the updated representation of $v_i$ in $l$-th layer. This iterative process enables nodes to integrate information from neighborhoods, capturing structural and feature patterns.

**Graph-aware and Graph-agnostic Models**   Neural networks that aggregate neighbors based on graph structure are called graph-aware models, typically paired with a graph-agnostic one. For

Figure 1: Overview of our LEMP4HG. (a) Illustration of embedding shift after message passing; (b) Heuristic definition to measure how much node pair suffer from message passing. Our pipeline includes three parts. (c) Initially, we finetune SLM for textual encoding with MLP as classifier; (d) Every $\mathcal{I}$ epochs, we select edges by MVRD to query LM for connection analysis; (e) Each epoch, we synthesize all encoded analysis and paired node texts to form enhanced messages for GNN training.

example, removing the neighboring aggregation from a 2-layer GCN reduces it to a 2-layer MLP.

$$\sigma(\hat{A}_{sym} \cdot \sigma(\hat{A}_{sym} X W_0) W_1) \xrightarrow{w/o\ AGGR} \sigma(\sigma(X W_0) W_1) \qquad (2)$$

where $\hat{A}_{sym} = \tilde{D}^{-1/2} \tilde{A} \tilde{D}^{-1/2}$, $\tilde{A} \equiv A + I$ and $\tilde{D} \equiv D + I$. In graph $G$, $A$ is adjacency matrix, $D$ is diagonal degree matrix and $I$ is identity matrix. Besides, $\sigma$ is activation function, e.g. ReLU.

**Homophily Metrics** There are many metrics for evaluating the homophily or heterophily of graph datasets from different views, such as the relations between node labels, features, and graph structures. Among them, edge homophily $\mathcal{H}_{edge}$ and node homophily $\mathcal{H}_{node}$ are commonly used ones.

$$\mathcal{H}_{edge}(\mathcal{G}) = \frac{\left| \{e_{ij} | e_{ij} \in \mathcal{E}, y_i = y_j\} \right|}{|\mathcal{E}|}, \quad \mathcal{H}_{node}(\mathcal{G}) = \frac{1}{|\mathcal{V}|} \sum_{v_i \in \mathcal{V}} \frac{\left| \{v_j | v_j \in \mathcal{N}(v_i), y_j = y_i\} \right|}{d_i} \qquad (3)$$

where $d_i$ is the degree of node $v_i$. $\mathcal{H}_{edge}(\mathcal{G})$ indicates the proportion of edges connecting two nodes from the same class. $\mathcal{H}_{node}(\mathcal{G})$ measures the average of local homophily by label-edge consistency.

## 3.2 OVERALL FRAMEWORK

Figure 1 is the overview of our proposed LEMP4HG. It illustrates our focus on embedding shift after message passing, heuristic definition and LM-enhanced pipeline. Initially, we use MLP as classifier to finetune SLM for text encoding. Every $\mathcal{I}$ epochs, we select edges to query LM guided by heuristic MVRD. Each epoch, we train GNN with all the available synthesized LM-enhanced messages.

## 3.3 LM-ENHANCED MESSAGE PASSING MECHANISM

In heterophilic regions, traditional message passing inevitably fuse contradictory signals between connected nodes, leading to suboptimal patterns learned by model. To address this, we propose a LM-Enhanced Message Passing (LEMP) mechanism, which can be summarized into the following three stages: LM Message Generation, Discriminative Message Synthesis, and Message Passing.

**LM Message Generation.** We design prompts $\pi$ (detailed in Appendix J) to query LM $\Psi_{LM}$ for the connection analysis of node pair $(v_i, v_j)$ with their associated texts $t_i$ and $t_j$. The response $t_{ij}$ is then encoded by finetuned SLM $\Phi_{SLM}$ as the preliminary message $h_{ij}$ for the subsequent process.

$$h_{ij} = \Phi_{SLM} \circ \Psi_{LM}(t_i, t_j; \pi), \quad \forall e_{ij} \in \mathcal{E} \text{ and } i \neq j \qquad (4)$$

**Discriminative Message Synthesis** To lower the cost of analysis generation by LM, node pair $(v_i, v_j)$ and $(v_j, v_i)$ share the same preliminary message, i.e. $\boldsymbol{h}_{ij} \equiv \boldsymbol{h}_{ji}$. However, the static nature of preliminary messages may hinder long-range neighbor aggregation, which relies on iteratively updated node representations in traditional message passing. Moreover, connection analysis may differ in semantic form from node texts, leading to misalignment in their encoded embeddings and noise introduction. To address these, we introduce a discriminative gating mechanism to fuse preliminary messages with source and target node embeddings, yielding final LM-enhanced messages.

$$\boldsymbol{\alpha}_{ij}^l = \sigma \left( \left[ \boldsymbol{h}_i^{l-1} \,\|\, \boldsymbol{h}_{ij} \,\|\, \boldsymbol{h}_j^{l-1} \right] \boldsymbol{W}_{gate} \right) \tag{5}$$

$$\boldsymbol{m}_{ij}^l = \beta \boldsymbol{h}_{ij} + (1 - \beta) \left[ \boldsymbol{\alpha}_{ij}^l \odot \boldsymbol{h}_i^{l-1} + (\mathbf{1} - \boldsymbol{\alpha}_{ij}^l) \odot \boldsymbol{h}_j^{l-1} \right] \tag{6}$$

where $\sigma$ is an activation function (e.g. Sigmoid), and $\boldsymbol{W}_{gate}$ is a trainable weight matrix. For node pair $(v_i, v_j)$, we concatenate $\boldsymbol{h}_i^{l-1}$ and $\boldsymbol{h}_j^{l-1}$ with $\boldsymbol{h}_{ij}$ to compute the gate weight $\boldsymbol{\alpha}_{ij}^l$, which together with a hyperparameter $\beta$ controls their contributions to the fused message $\boldsymbol{m}_{ij}^l$.

**Message Passing** Unlike the classic message passing mechanism as shown in Preliminary 3.1, we employ LM-enhanced message $\boldsymbol{m}_{ij}^l$ to substitute the neighbor representation $\boldsymbol{h}_j^{l-1}$ for propagation.

$$\boldsymbol{h}_j^l = UPDATE(\boldsymbol{h}_j^{l-1}, \{\boldsymbol{m}_{ij} | v_i \in \mathcal{N}(v_j)\}) = \sigma(\hat{\boldsymbol{A}}_{sym}^{jj} \cdot \boldsymbol{h}_j^{l-1} + \sum_{v_i \in \mathcal{N}(v_j)} \hat{\boldsymbol{A}}_{sym}^{ij} \cdot \boldsymbol{m}_{ij}^l) \tag{7}$$

where $\sigma$ includes batch normalization, activation functions (e.g. ReLU), and dropout.

### 3.4 HEURISTIC FOR EVALUATING MESSAGE PASSING

**Assumptions.** *(1) Nodes with similar features are more likely to share the same category labels; (2) A node's representation and its classification confidence tend to be more reliable when it lies nearer to its embedding cluster center; (3) GNNs favor mild smoothing, while excessive contraction of representations between heterophilic node pairs usually indicates representation distortion.*

Building on these assumptions, we propose **MVRD** (**M**odulated **V**ariation of **R**eliable **D**ifference) as a heuristic to evaluate the effect of message passing from the perspective of paired node embedding contraction, capturing representation distortion commonly arise in heterophilic regions and suppress benign convergence typically in homophilic regions. The specific calculation steps are as below.

**Reliable Difference** To evaluate the difference between the connected nodes in the embedding space reliably, we firstly cluster the node representations in the embedding space in a semi-supervised way (detailed in Appendix F.4). For each node pair $(v_i, v_j)$ with $e_{ij} \in \mathcal{E}$, we compute their euclidean distance $d_{ij}$ and their respective distances to cluster centers, $d_i$ and $d_j$. Then, the reliable difference $RD_{ij}$ between node $v_i$ and $v_j$ in the embedding space can be measured as below.

$$RD_{ij} = \frac{\sigma(\gamma \cdot d_{ij})}{\sigma(d_i + d_j)} = \frac{\sigma(\gamma \cdot \|\boldsymbol{h}_i - \boldsymbol{h}_j\|)}{\sigma(\|\boldsymbol{h}_i - \boldsymbol{c}_{\hat{y}_i}\| + \|\boldsymbol{h}_j - \boldsymbol{c}_{\hat{y}_j}\|)}, \text{ where } \boldsymbol{c}_k = \frac{1}{|\{l : \hat{y}_l = k\}|} \sum_{\hat{y}_l = k} \boldsymbol{h}_l \tag{8}$$

where $\hat{y}_l$ is cluster label of node $v_l$, $\boldsymbol{c}_k$ is center embedding of $k$-th cluster, $\sigma$ is an activation function (e.g. Sigmoid) and $\gamma > 0$ balances the influence of two types of distances. Smaller $d_i$ and $d_j$ imply more reliable node representation, thus a more reliable difference measure between $\boldsymbol{h}_i$ and $\boldsymbol{h}_j$. $RD_{ij}$ is strictly increasing w.r.t. $d_{ij}$, and strictly decreasing w.r.t. $d_i$ and $d_j$, proved in Appendix H.1.

**Variation** Representation distortion arise when message passing between dissimilar nodes draws their embeddings closer and away from correct classification regions. Thus, we compute variation of reliable difference after message passing to measure the effect. With embedding space **b**efore $l_b$-th and **a**fter $l_a$-th layer aggregation as $\boldsymbol{H}_b^{l_b} = \sigma(\boldsymbol{H}^{l_b-1}\boldsymbol{W} + \boldsymbol{b})$ and $\boldsymbol{H}_a^{l_a} = \sigma(\hat{\boldsymbol{A}}_{sym}(\boldsymbol{H}^{l_a-1}\boldsymbol{W} + \boldsymbol{b}))$, we compute $RD_{ij}^{l_b,b}$ and $RD_{ij}^{l_a,a}$ for each connected pair $(v_i, v_j)$, and define the variation as below.

$$VRD_{ij}^{l_b,l_a} = RD_{ij}^{l_b,b} - RD_{ij}^{l_a,a}, \quad l_a, l_b \in \{1, 2, ..., N_l\} \text{ and } l_b \le l_a \tag{9}$$

where $N_l$ is the total number of message passing layers. In this paper, we set $l_a = l_b = 1$, focus on the effect of the first-round message passing. Thus, we abbreviate the notations as $RD_{ij}^b$, $RD_{ij}^a$ and $VRD_{ij}$. In summary, $VRD_{ij}$ measures the decline of reliable difference after one-layer message passing. A higher $VRD_{ij}$ indicates a greater negative effect of message passing between $(v_i, v_j)$.

Table 1: Categorization of TAG datasets. H-Cat is based on $\mathcal{H}_{node}$ and $\mathcal{H}_{edge}$, while MP-Cat reflects the performance shift after message passing. Specifically, datasets exhibiting performance decline after message passing are classified as malignant, improvements as benign, and others as ambiguous.

| H-Cat. | Datasets | MP-Cat. | $\mathcal{H}_{node}$ | $\mathcal{H}_{edge}$ | 2-MLP | 4-MLP | 2-GCN | 4-GCN |
|---|---|---|---|---|---|---|---|---|
| Heterophily | Cornell | Malignant | 0.1155 | 0.1241 | **0.8654 ± 0.0674** | **0.8333 ± 0.0948** | 0.6474 ± 0.0529 | 0.5321 ± 0.1347 |
| | Texas | | 0.0661 | 0.0643 | **0.8462 ± 0.0000** | **0.8205 ± 0.0363** | 0.6090 ± 0.0706 | 0.5705 ± 0.0245 |
| | Washington | | 0.1610 | 0.1507 | **0.8404 ± 0.0662** | **0.8511 ± 0.0796** | 0.6543 ± 0.0268 | 0.6383 ± 0.0174 |
| | Wisconsin | | 0.1609 | 0.1808 | **0.8796 ± 0.0685** | **0.8981 ± 0.0717** | 0.5972 ± 0.1073 | 0.5324 ± 0.1204 |
| | arxiv23 | | 0.2966 | 0.6443 | **0.7811 ± 0.0035** | **0.7774 ± 0.0028** | 0.7781 ± 0.0021 | 0.7705 ± 0.0017 |
| | Children | | 0.4559 | 0.4043 | **0.6199 ± 0.0071** | **0.6136 ± 0.0064** | 0.6054 ± 0.0085 | 0.5880 ± 0.0192 |
| | Amazon | Benign | 0.3757 | 0.3804 | 0.4275 ± 0.0087 | 0.4346 ± 0.0224 | **0.4543 ± 0.0118** | **0.4495 ± 0.0052** |
| Homophily | Pubmed | Malignant | 0.7924 | 0.8024 | **0.9471 ± 0.0043** | **0.9473 ± 0.0036** | 0.9349 ± 0.0029 | 0.9326 ± 0.0011 |
| | History | | 0.7805 | 0.6398 | **0.8616 ± 0.0052** | **0.8554 ± 0.0059** | 0.8540 ± 0.0060 | 0.8483 ± 0.0053 |
| | Cora | Benign | 0.8252 | 0.8100 | 0.8034 ± 0.0161 | 0.7947 ± 0.0244 | **0.8743 ± 0.0190** | **0.8840 ± 0.0086** |
| | citeseer | | 0.7440 | 0.7841 | 0.7371 ± 0.0116 | 0.7351 ± 0.0095 | **0.7853 ± 0.0128** | **0.7857 ± 0.0167** |
| | Photo | | 0.7850 | 0.7351 | 0.7124 ± 0.0006 | 0.7133 ± 0.0020 | **0.8541 ± 0.0065** | **0.8577 ± 0.0023** |
| | Computers | | 0.8528 | 0.8228 | 0.6073 ± 0.0044 | 0.6042 ± 0.0016 | **0.8710 ± 0.0028** | **0.8806 ± 0.0024** |
| | Fitness | | 0.9000 | 0.8980 | 0.8969 ± 0.0010 | 0.8958 ± 0.0025 | **0.9277 ± 0.0002** | **0.9286 ± 0.0004** |
| | wikics | Ambiguous | 0.6579 | 0.6543 | 0.8597 ± 0.0060 | **0.8599 ± 0.0046** | **0.8672 ± 0.0073** | 0.8549 ± 0.0013 |
| | tolokers | | 0.6344 | 0.5945 | **0.7793 ± 0.0096** | 0.7824 ± 0.0044 | 0.7783 ± 0.0072 | **0.7848 ± 0.0038** |

**Modulation**   While $VRD_{ij}$ tends to increase with higher $RD_{ij}^b$ and lower $RD_{ij}^a$, an extremely small $d_{ij}^a$—indicating that $v_i$ and $v_j$ become highly similar after aggregation—often reflects effective neighbor aggregation in homophilic regions rather than representation distortion. To suppress the benign convergence and prevent overestimation of $VRD_{ij}$ in such case, we introduce a modulation:

$$MVRD_{ij} = \sigma\left(\eta \cdot d_{ij}^a\right) \cdot VRD_{ij} \tag{10}$$

where $\sigma$ denotes activation function (e.g. Sigmoid) and $\eta$ balances the influence of modulation.

### 3.5 ACTIVE LEARNING FOR EDGE SELECTION

To scale our LM-enhanced message passing for large graphs, it's impractical to enhance all edges with $O(E)$ complexity. Thus, we use an active learning strategy with heuristic MVRD to identify and enhance edges prone to suffer from message passing. However, active learning strategy struggle with unstable model weights and node representations in early training stages, leading to suboptimal edge selection. Thus, we introduce a weight-free auxiliary model for stable guidance. Specifically, a weight-free 2-layer GCN can be formulated as $\mathcal{M}^{wf} : \sigma(\hat{\boldsymbol{A}}_{sym} \cdot \sigma(\hat{\boldsymbol{A}}_{sym}\boldsymbol{X}))$, while its paired weight-based one is $\mathcal{M}^{wb} : \sigma(\hat{\boldsymbol{A}}_{sym} \cdot \sigma(\hat{\boldsymbol{A}}_{sym}\boldsymbol{X}\boldsymbol{W}_0)\boldsymbol{W}_1)$. We then compute $MVRD_{ij}^{wf}$ and $MVRD_{ij}^{wb}$ as Equation 8-10 and introduce a time-sensitive weight $\lambda$ to fuse them as below:

$$\lambda = \omega \cdot cos\left(\frac{\pi \cdot n_e}{N_e}\right) + \phi, \text{ with } N_e = k\frac{\mathcal{B}}{\mathcal{I}} \tag{11}$$

$$MVRD_{ij} = \lambda \cdot MVRD_{ij}^{wf} + (1 - \lambda) \cdot MVRD_{ij}^{wb} \tag{12}$$

where $n_e$ is current training epoch, $\mathcal{B}$, $\mathcal{I}$ and $k$ are budget, epoch interval and batch size for query. During training, we select top-$k$ edges with highest $MVRD$ scores every $\mathcal{I}$ epochs to query LM for connection analysis to enhance messages. Training stops at budget exhaustion or patience limit.

## 4 EXPERIMENTS

### 4.1 EXPERIMENT SETUP

**Datasets**   Since commonly used datasets for heterophilic graph tasks often lack raw textual information, we collect 16 publicly available raw text datasets as recent studies (Liu et al., 2023; Yan et al., 2023). Statistical details are presented in Table 6, with comprehensive descriptions in Appendix C. For each dataset, we compute node and edge homophily scores $\mathcal{H}_{node}$ and $\mathcal{H}_{edge}$, and evaluate using both graph-aware (GCN) and graph-agnostic (MLP) models with 2- and 4-layer configurations. Results are summarized in Table 1. Datasets are categorized based on homophily metrics and the performance shift after message passing. Following Luan et al.(Luan et al., 2024), datasets exhibiting performance decline after message passing are classified as malignant, improvements as benign, and others as ambiguous. Notably, we observe that even datasets deemed homophily by $\mathcal{H}_{node}$ and $\mathcal{H}_{edge}$ can exhibit malignant or ambiguous behavior, extending prior findings and emphasizing the importance of identifying challenging datasets based on performance shifts after message passing rather than solely on homophily metrics.

Table 2: Evaluation of our LEMP4HG and baselines on various text-attributed graphs. "OOT" and "OOM" denote runtime or memory limits failures. "Down" indicates negative impact of our method. "+T" denotes enhanced by TAPE. **Bold** numbers indicate optimal average performance ranking.

| Models | Heterophilic Graph | | | | | | | Homophilic Graph | | | | | | | | | Rank | |
|---|---|---|---|---|---|---|---|---|---|---|---|---|---|---|---|---|---|---|
| | Cornell | Texas | Wash. | Wis. | arxiv23 | Child | Amazon | Pubmed | History | Cora | citeseer | Photo | Comp. | Fitness | wikics | tolokers | w/ $f_4$ | w/o $f_4$ |
| MLP | 0.8654 | 0.8462 | 0.8404 | 0.8796 | 0.7811 | 0.6199 | 0.4275 | 0.9471 | 0.8616 | 0.8034 | 0.7371 | 0.7121 | 0.6065 | 0.8969 | 0.8597 | 0.7793 | 13.38 | 15.33 |
| GCN | 0.6346 | 0.6026 | 0.6596 | 0.5972 | 0.7785 | 0.6083 | 0.4558 | 0.9354 | 0.8559 | 0.8762 | 0.7853 | 0.8563 | 0.8735 | 0.9282 | 0.8700 | 0.7820 | 12.06 | 9.08 |
| SAGE | 0.8269 | 0.8269 | 0.8564 | 0.8935 | 0.7861 | 0.6245 | 0.4648 | 0.9475 | 0.8649 | 0.8531 | 0.7813 | 0.8518 | 0.8727 | 0.9240 | 0.8771 | 0.7885 | 6.81 | 6.42 |
| GAT | 0.4808 | 0.5962 | 0.5532 | 0.4769 | 0.7622 | 0.5824 | 0.4520 | 0.8875 | 0.8441 | 0.8725 | 0.7841 | 0.8545 | 0.8738 | 0.9261 | 0.8533 | 0.7821 | 14.50 | 12.00 |
| RevGAT | 0.8397 | 0.8205 | 0.8777 | 0.8935 | 0.7798 | 0.6195 | 0.4590 | 0.9484 | 0.8645 | 0.8085 | 0.7551 | 0.7839 | 0.7597 | 0.9083 | 0.8665 | 0.7968 | 9.69 | 10.67 |
| Cheby | 0.8654 | 0.8462 | 0.8404 | 0.8796 | 0.7811 | 0.6199 | 0.4275 | 0.9471 | 0.8616 | 0.8034 | 0.7371 | 0.7114 | 0.6045 | 0.8969 | 0.8597 | 0.7793 | 13.50 | 15.50 |
| JKNet | 0.6603 | 0.6410 | 0.7181 | 0.6667 | 0.7774 | 0.6031 | 0.4551 | 0.9314 | 0.8537 | 0.8821 | 0.7845 | 0.8545 | 0.8739 | 0.9282 | 0.8629 | 0.7838 | 12.00 | 9.58 |
| APPNP | 0.6474 | 0.6538 | 0.7500 | 0.6250 | 0.7762 | 0.6241 | 0.4534 | 0.9066 | 0.8569 | 0.8821 | 0.7927 | 0.8446 | 0.8647 | 0.9279 | 0.8754 | 0.7809 | 12.13 | 9.75 |
| H2GCN | 0.6795 | 0.7244 | 0.8138 | 0.7639 | 0.7761 | 0.6126 | 0.4071 | 0.9473 | 0.8383 | 0.8324 | 0.7712 | 0.8441 | 0.8632 | 0.9178 | 0.8660 | 0.7815 | 14.56 | 13.67 |
| GCNII | 0.8013 | 0.8013 | 0.8564 | 0.8750 | 0.7832 | 0.6223 | 0.4429 | 0.9483 | 0.8630 | 0.8352 | 0.7441 | 0.8493 | 0.8736 | 0.9137 | 0.8674 | 0.7861 | 9.81 | 9.08 |
| FAGCN | 0.7051 | 0.7949 | 0.7074 | 0.8102 | 0.7446 | 0.6287 | 0.4319 | 0.8859 | 0.7784 | 0.8191 | 0.7555 | 0.8080 | 0.7216 | 0.7790 | 0.8655 | 0.7812 | 15.94 | 15.42 |
| GPR | 0.8269 | 0.8333 | 0.8564 | 0.8796 | 0.7815 | 0.6316 | 0.4554 | 0.9470 | 0.8583 | 0.8794 | 0.7861 | 0.8467 | 0.8728 | 0.9277 | 0.8755 | 0.7813 | 8.25 | 8.00 |
| Jacobi | 0.7756 | 0.7692 | 0.7766 | 0.8426 | 0.7153 | 0.5981 | 0.4554 | 0.9473 | 0.8543 | 0.8734 | 0.7810 | 0.8432 | 0.8610 | 0.9238 | 0.8778 | 0.7817 | 12.63 | 11.33 |
| GBK | 0.8333 | 0.8397 | 0.8085 | 0.8889 | 0.7617 | 0.4961 | 0.4274 | 0.9476 | 0.8403 | 0.8250 | 0.7649 | 0.7659 | 0.6954 | 0.8771 | 0.8723 | 0.7799 | 14.25 | 15.58 |
| OGNN | 0.8462 | 0.8397 | 0.8564 | 0.8981 | 0.7820 | 0.6250 | 0.4366 | 0.9487 | 0.8633 | 0.8066 | 0.7504 | 0.7914 | 0.7693 | 0.9095 | 0.8684 | 0.7803 | 10.00 | 11.50 |
| SEGSL | 0.8333 | 0.8590 | 0.8564 | 0.9028 | OOT | OOT | OOT | OOT | OOT | 0.8191 | 0.7680 | OOT | OOT | OOT | OOT | OOT | 7.67 | 14.00 |
| Disam | 0.8462 | 0.8141 | 0.8404 | 0.8704 | 0.7801 | OOM | 0.4410 | 0.9476 | 0.8604 | 0.8103 | 0.7343 | OOM | OOM | OOM | 0.8651 | 0.7835 | 12.67 | 13.25 |
| SATA | 0.8141 | 0.8077 | 0.8457 | 0.8935 | OOM | OOM | 0.4237 | 0.9453 | OOM | 0.8043 | 0.7339 | OOM | OOM | OOM | 0.8602 | 0.7815 | 15.30 | 18.17 |
| SAGE+T | 0.8718 | 0.8526 | 0.8670 | 0.8889 | 0.8023 | 0.6316 | 0.4639 | 0.9480 | 0.8677 | 0.8771 | 0.7837 | 0.8587 | 0.8733 | 0.9315 | 0.8823 | 0.7848 | 4.00 | 4.13 |
| RevGAT+T | 0.8846 | 0.8590 | 0.8777 | 0.9074 | 0.7995 | 0.6285 | 0.4722 | 0.9480 | 0.8664 | 0.8439 | 0.7774 | 0.8002 | 0.7640 | 0.9215 | 0.8824 | 0.7991 | 5.06 | 5.32 |
| LEMP | 0.8526 | 0.8269 | 0.8564 | 0.8981 | 0.7853 | 0.6137 | 0.4590 | 0.9485 | 0.8590 | 0.8803 | 0.7888 | Down | Down | Down | 0.8729 | 0.7867 | 6.00 | 6.00 |
| LEMP+T | 0.8654 | 0.8590 | 0.8777 | 0.8565 | 0.8003 | 0.6179 | 0.4675 | 0.9484 | 0.8662 | 0.8826 | 0.7943 | 0.8591 | 0.8729 | 0.9303 | 0.8825 | 0.7897 | **3.56** | **3.60** |

**Baselines** We compare our LEMP4HG against four categories of baselines: MLP, classic GNNs (GCN, SAGE, GAT, RevGAT, GCN-Cheby, JKNet, APPNP), heterophily-specific GNNs (H2GCN, GCNII, FAGCN, GPRGNN, JacobiConv, GBK-GNN, OGNN, SEGSL, DisamGCL, GNN-SATA), LM-enhanced GNNs (SAGE- and RevGAT-backboned TAPE, LLM4HeG). Specifically, the results of LLM4HeG is presented in Appendix I.2. Details of datasets are shown in Appendix D.

**Implementation** We adopt Qwen-turbo as LM to generate connection analysis via API calls and DeBERTa-base (He et al., 2021b) as SLM to encode texts. Following the common practice, we randomly split nodes into train, validation and test sets as Table 6, where all experiments are performed with 4 runs and reported as average results with standard deviation provided in Appendix I.

## 4.2 MAIN RESULTS

We evaluate our LEMP4HG on 16 homogeneous text-attributed graph (TAG) datasets, comparing it against four categories of baselines in Table 2. Compared with backbone GCN, LEMP4HG achieves performance gains on 13 out of 16 datasets. The exceptions (Photo, Comp., Fitness) are likely due to their large graph size, high homophily level and benign message passing effect, which reduce insufficient LM-enhanced messages to noise. We report average performance rankings with and without $f_4$, which includes four small datasets (Cornell, Texas, Washington, and Wisconsin) known for their unstable behavior. From the results, we find LEMP4HG consistently outperforms MLP, classical GNNs, and heterophily-specific GNNs. Although TAPE-enhanced SAGE (TAPE+S) and RevGAT (TAPE+R) rank better, their advantage comes from the ensemble of three separate models. TAPE also boost LEMP4HG (LEMP+T) and rectify the three instances of performance decline.

**Fair Evaluation** To fairly evaluate graph models across diverse scenarios, we report their distribution of the average ranking performance on five dataset categories (heterophily, homophily, malignant, benign and ambiguous) with boxplot in Figure 2. Results show that most heterophily-specific baselines fail to maintain performance across datasets with varying homophily levels and message passing effects, whereas LEMP4HG achieves more robust and superior performance across all categories.

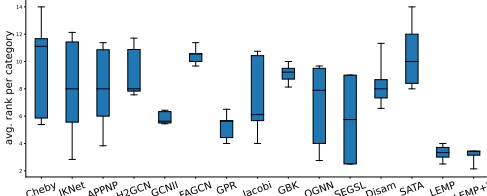

Figure 2: Rank distribution on 5 dataset categories. Lower the box, more robust the model.

Table 3: Ablation studies: heuristic definition and message synthesis. **Bold** indicates the optimal performance, while underlined ones the runner-up. ”\” indicates consistency with no ablation.

| Ablation | Varient | Cornell | Texas | Wash. | Wis. | arxiv23 | Child | Amazon | Pubmed | History | Cora | citeseer | wikics | tolokers |
|---|---|---|---|---|---|---|---|---|---|---|---|---|---|---|
| Heuristic Definition | **MVRD** | 0.8526 | 0.8269 | 0.8564 | 0.8981 | **0.7853** | **0.6160** | **0.4590** | **0.9485** | **0.8599** | 0.8803 | 0.7888 | **0.8768** | 0.7867 |
| | VRD | \ | \ | \ | \ | 0.7811 | 0.6116 | 0.4578 | 0.9469 | 0.8579 | **0.8821** | 0.7861 | 0.8721 | 0.7861 |
| | featDiff | \ | \ | \ | \ | 0.7829 | 0.6113 | 0.4577 | 0.9453 | 0.8576 | 0.8752 | **0.7896** | 0.8680 | **0.7876** |
| Message Synthesis | **w/ mn** | **0.8526** | **0.8269** | **0.8564** | **0.8981** | **0.7853** | 0.6160 | 0.4590 | **0.9485** | 0.8599 | 0.8803 | 0.7888 | **0.8768** | 0.7867 |
| | w/o m | 0.7628 | 0.7756 | 0.7713 | 0.8519 | 0.7852 | **0.6237** | 0.4566 | 0.9471 | **0.8626** | 0.8439 | 0.7633 | 0.8725 | 0.7843 |
| | w/o n | 0.7756 | 0.8013 | 0.7394 | 0.8472 | 0.7814 | 0.6119 | **0.4602** | 0.9472 | 0.8582 | **0.8821** | 0.7880 | 0.8731 | 0.7861 |
| | w/o mn | 0.5577 | 0.6026 | 0.7074 | 0.5972 | 0.7777 | 0.6094 | 0.4502 | 0.9364 | 0.8509 | 0.8435 | **0.7888** | 0.8694 | 0.7809 |

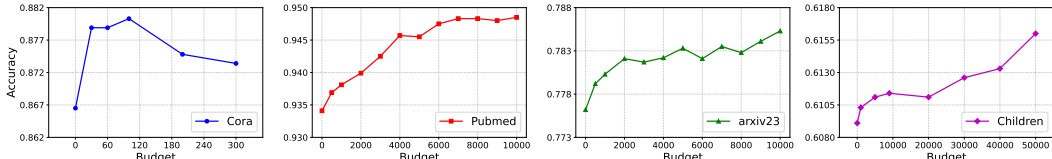

Figure 3: Scalability study on Cora, Pubmed, arxiv23 and Children: accuracy v.s. budget

## 4.3 ABLATION STUDY

We conduct an ablation study on heuristic definition and message synthesis. For heuristic, we compare MVRD, VRD (no modulation), and FeatDiff, which selects top-k edges by feature difference and consumes the full budget $\mathcal{B}$ initially. Notably, all edges of small datasets $f_4$ are enhanced by LM, resulting in no variation across heuristics. For messsage synthesis, our LEMP4HG fuse preliminary messages with paired node textual embeddings via a gating mechanism. We evaluate three variants: w/o m (no preliminary messages), w/o n (no node textual embeddings), and w/o mn (GCN backbone). Table 3 presents results averaged over four runs, demonstrating the effectiveness of ours. Specifically, without modulation, VRD falsely attributes the benign convergence in homophilic regions to representation distortion, introducing potential noise by LM-enhanced messages. FeatDiff, a naive method based on predefined heterophily metrics fail to effectively identifying node pairs that suffer most from message passing. For message synthesis, w/o m results in suboptimal performance for limited semantics and information sources, while w/o n suffer from the absence of multi-hop information and misalignment with node embedding pattern when aggregation.

## 4.4 SCALABILITY

To evaluate the scalability of LEMP4HG, we vary the budget **B** for LM query and observe that the scaling-up rule generally holds across most datasets. Figure 3 presents the results for Cora, Pubmed, arxiv23 and Children, while details of the remaining datasets are provided in Appendix I.3. In the figure, four datasets all exhibit accuracy improvements as the budget increases, with gains of 1.38%, 1.44%, 0.91% and 0.69% respectively. However, Cora experiences a performance peak followed by a decline, whereas the others show steadily increasing trend with diminishing returns, which results from the differences in dataset size and homophilic characteristics. Specifically, without a sufficient budget, introducing LM-enhanced messages in larger graph datasets may introduce additional noise instead of improvements, especially in homophilic and message passing-benign graph regions.

**Budget Allocation Guidelines** Based on empirical scalability analysis, we propose the following budget allocation recommendations: (1) For homophilic datasets where MP-Cat is benign, allocate $\mathcal{B}^* \approx 5\% \cdot |\mathcal{E}|$ for small graphs (e.g. Cora), while $\mathcal{B}^* = 0$ for large graphs (e.g. Photo); (2) For other dataset categories, including heterophilic garphs and homophilic graphs with MP-Cat as malignant or ambiguous, performance exhibits a strong correlation with $\mathcal{B}$. In these cases, progressively increasing the budget is recommended until performance reaches saturation.

## 4.5 CASE STUDY

Both Pubmed and Cornell show malignant outcomes after message passing, with Pubmed displaying homophily while Cornell heterophily. We conduct the case study in the following two aspects.

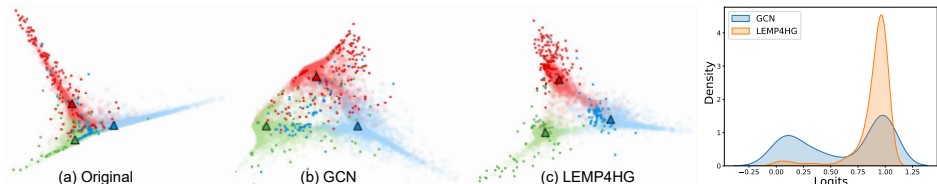

Figure 4: (left) Embedding space before and after message passing. (right) Logits distribution.

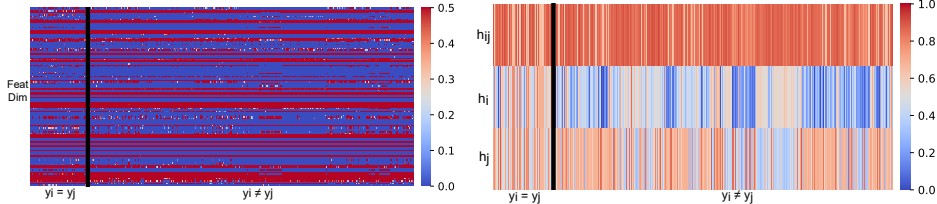

Figure 5: (left) Gate vector that balances the contribution of source and target node embeddings. (right) Similarity matrix between synthesized message $m_{ij}$ and preliminary one $h_{ij}$, source and target node embedding $h_i$, $h_j$. The vertical line separates all node pairs into $y_i = y_j$ and $y_i \neq y_j$.

**MVRD-guided Edge Selection**    To investigate the nodes characterized by MVRD, we select top-300 nodes most frequently involved in the queried node pairs during our experiment on Pubmed with budget $\mathcal{B} = 10,000$. In Figure 4 (left), we highlight these nodes in embedding spaces derived from the original node features and hidden layer of the trained GCN and LEMP4HG. Different colors denote different classes, with intensity indicating local density. We observe that both GCN and LEMP4HG form clearer class cluster than the original features. However, these nodes represented by GCN often drift into ambiguous intersection regions, while our LEMP4HG refines their representations, placing them in regions that favor correct classification. Furthermore, Table 4 (right) presents the distribution of normalized prediction logits for correct labels of these nodes with kernel density estimation, indicating that LEMP4HG notably enhances classification on these nodes.

**Message Synthesis**    We analyze the message synthesis of dataset Cornell with discrimination on paired node labels ($y_i = y_j$ or $y_i \neq y_j$). In Figure 5 (left), we visualize the gate vector $\boldsymbol{\alpha}_{ij}^1$ in Equation 6 with $l = 1$. It demonstrates that source $h_i$ and target $h_j$ node embedding are regularly integrated into preliminary message $h_{ij}$ in the dimensional-level, potentially aligning with semantic structure of LM-generated connection analysis. Then, we illustrate the cosine similarity between synthesized message $m_{ij}$ and preliminary message $h_{ij}$, source nodes features $h_i$ and target ones $h_j$ in the (right). We observe that preliminary message $h_{ij}$ consistently contributes most, while the source node embedding $h_i$ contributes more in homophilic regions ($y_i = y_j$) than heterophilic ones ($y_i \neq y_j$), conforming that message passing from source node to target one benefits from homophily.

# 5 CONCLUSION

In this paper, we propose a language model (LM)-enhanced message passing approach for heterophilic graph learning (LEMP4HG). For text-attributed graph, we leverage a finetuned small language model (SLM) to encode textual content, which unlock the semantic potential for graph-based tasks. To further integrate LM for heterophilic scenarios, we provide another LM with the associated texts of node pairs to generate their connection analysis, which are encoded and fused with source and target node textual embeddings. The synthesized messages are semantically enriched and balanced with paired node representations dynamically for propagation, mitigating contradictory signals in heterophilic regions. Furthermore, we introduce an active learning strategy guided by our heuristic MVRD (Modulated Variation of Reliable Distance), selectively enhancing node pairs suffer most from message passing, reducing the cost of analysis generation and side effects on homophilic regions. Extensive experiments demonstrate that LEMP4HG excels on heterophilic graphs and performs robustly on homophilic ones, with a simple GCN backbone and a practical budget.

## A  ETHICS STATEMENT

This research complies with the ICLR Code of Ethics. Our work did not involve human participants or animal testing. All datasets were obtained in accordance with applicable usage policies, with privacy protection as a priority. We implemented measures to prevent biases and discriminatory results, and no personally identifiable information was utilized. The study design avoids any potential privacy or security issues, upholding our commitment to research integrity.

## B  REPRODUCIBILITY STATEMENT

To facilitate the reproduction of our results, we will release the complete codebase and cached LM outputs upon acceptance. Detailed experimental configurations, including hardware specifications, software environment, and hyperparameter settings, are provided in Section F. The core methods, such as SLM fine-tuning, GNN training protocols, and heuristic definitions for edge selection, are comprehensively described in Sections 3, Appendix F and their respective subsections.

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

# A ADVANTAGES OF OUR LEMP4HG

## A.1 ROBUSTNESS

**Performance robustness across different dataset categories**: A robust heterophily-specific model should excel on heterophilic graphs while maintaining at least parity on homophilic ones, while many heterophiliy-specific GNN sacrifice their performance on homophilic graphs to achieve success on heterophilic ones. As the pattern for graph datasets is more complex than "homophily wins, heterophily loses", we classify graph datasets into heterophilic, homophilic, as well as malignant, benign and ambiguous for a comprehensive evaluation. The results shown by Table 4 demonstrates the robustness of our LEMP4HG with performance superiority across all categories of datasets.

Table 4: Average ranks by dataset category. **Bold**: top-3 models; -: frequent OOM.

| Dataset | Cheby | JKNet | APPNP | H2GCN | GCNII | FAGCN | GPR | Jacobi | GBK | OGNN | SEGSL | Disam | SATA | LEMP | LEMP+T |
|---|---|---|---|---|---|---|---|---|---|---|---|---|---|---|---|
| heterophilic | 5.86 | 11.43 | 10.86 | 11.71 | 6.43 | 10.57 | 4.43 | 10.43 | 8.71 | **4.00** | - | 7.33 | 8.40 | **3.71** | **3.14** |
| homophilic | 11.11 | 5.56 | 6.00 | 7.56 | **5.44** | 10.56 | 5.67 | 6.11 | 9.22 | 7.89 | 9.00 | 8.67 | 12.00 | **3.33** | **3.43** |
| malignant | 5.38 | 12.13 | 11.38 | 10.88 | 5.63 | 11.38 | 5.63 | 10.75 | 8.13 | **2.75** | - | 6.57 | 8.00 | **4.00** | **3.43** |
| benign | 11.67 | **2.83** | 3.83 | 7.83 | 6.33 | 9.67 | 4.00 | 5.67 | 10.00 | 9.67 | 9.00 | 11.33 | 14.00 | **3.00** | **3.43** |
| ambiguous | 14.00 | 8.00 | 8.00 | 8.00 | 5.50 | 10.00 | 6.50 | **4.00** | 9.50 | 9.50 | – | 8.00 | 10.00 | **2.50** | **2.14** |

**Robustness to hyper-parameter**: Many superior performances of existing GNNs often come from extensive hyper-parameter searches across large configuration spaces, e.g. JacobiConv and GNN-SATA included in our comparative experiments. While, built upon a **simple 2-layer GCN backbone**, LEMP4HG performs well with default hyper-parameters across datasets, requiring little tuning thanks to its hyper-parameter robustness. The main hyper-parameters for tuning in LEMP4HG are only $\gamma$ in Equation 8 and $\eta$ in Equation 10 for heuristic design, illustrated in Appendix G.

**Robustness to language model outputs**: Our LEMP4HG exhibits strong robustness to the variation in LM outputs compared to LLM4HeG, which is high sensitivity to output heterogeneity—especially in ways that directly impact the edge reweighting. Our approach avoids this issue through a key strategy: instead of relying on exact connection analyses, we encode and integrate LM outputs into message representations with paired node features. This integration mitigates variability and effectively bridges semantic gaps, ensuring stable performance across diverse LM responses. We conducted experiments using three increasingly capable variants of the Qwen family across four datasets with the same budget as shown in Table 5, indicating that simply employing a more advanced and computationally expensive LM does not yield significant performance gains. Given this stability, a more economical and faster LM variant (e.g., qwen-turbo) is preferable in practical deployment, offering reduced latency and cost without sacrificing performance. This reinforces our method's robustness to LM variability and its suitability for real-world scenarios where efficiency is critical.

Table 5: Performance comparison with different Qwen LM variants. Model capability and API price rank as: qwen-max > qwen-plus > qwen-turbo, while response speed ranks in the reverse order.

| Dataset | qwen-turbo | qwen-plus | qwen-max |
|---|---|---|---|
| Cora | 0.8803 ± 0.0145 | 0.8812 ± 0.0164 | 0.8807 ± 0.0146 |
| citeseer | 0.7888 ± 0.0078 | 0.7898 ± 0.0122 | 0.7883 ± 0.0115 |
| Pubmed | 0.9485 ± 0.0042 | 0.9490 ± 0.0045 | 0.9497 ± 0.0046 |
| arxiv23 | 0.7853 ± 0.0026 | 0.7860 ± 0.0022 | 0.7868 ± 0.0031 |

## A.2 INTERPRETABILITY OF LANGUAGE MODEL-ENHANCED MESSAGE REPRESENTATION

In fact, there are some heterophily-specific GNN that edits graph structure by adding homophilic or removing heterphilic edges. In our view, these methods are compromises of traditional message-passing mechanisms in heterophilic scenarios. In the real-world, any interaction can bring information for deicision. The message passing between heterophilic node pairs is not inherently flawed; rather, the fundamental limitation stems from traditional message-passing mechanisms' reliance on propagating a node's own features as the message. A superior message representation should emerge as the product of bidirectional information interaction and reasoning between nodes, which can be realized by language model. **The integrated corpus in the reasoning process serve as augmented data that facilitate bridging the semantic gap between heterophilic nodes.**

## A.3 EFFICIENCY

In graph learning, edge-level complexity—which typically scales as O(E)—is often impractical for large graphs due to the sheer number of edges involved. To address this, we propose a heuristic-guided active learning strategy for edge-level message representation enhancement with LM, which selectively queries the paired nodes suffer most from message passing (i.e. most informative edges). This significantly improves feasibility and scalability of integrating LM signals into graph learning.

In contrast to prior works such as LLM4HeG, which rely on LLM fine-tuning and LLM-to-SLM distillation—both of which are resource-intensive—our LEMP4HG framework adopts a lightweight, API-based querying approach. This strategy reduces both computational overhead and deployment costs, **enabling immediate integration of LM capabilities without heavy training pipelines**.

## B REAL-WORLD APPLICABILITY

**LEMP4HG is suited for the scenarios** as our budget allocation guidelines presented in Section 4.4: heterophilic graphs, homophilic graphs with malignant message passing, and small to medium-sized homophilic graphs with benign message passing. For large homophilic graphs with benign message passing, some lightweight GNNs are already effective, making further enhancement unnecessary.

**For extremely large-scale graphs**, any enhancement based on LM inference is hard to scale. Our heuristic-guided active learning strategy for selective message enhancement has reduced the cost while retaining the benefits of semantic augmentation.

**For non-text-attributed graphs**, our LEMP4HG can work as long as informative textual representations can be constructed, as demonstrated by the tolokers dataset with numerical and categorical features converted via templates. Generalization to domains like molecular graphs remains a promising future direction.

## C DETAILS OF DATASETS

### C.1 STATISTICS

Table 6 have shown the statistics of our selected 16 datasets, including the domain of Academic Webpage, Biology Citation, Computer Science Citation, E-Commerce and Knowledge. Specifically, we count the edges by treating the graph as the undirected one and remove the self-loops. As for the split of these datasets, we mostly adhere to their original strategies (Chen et al., 2024a; Ramsundar et al., 2019; Liu et al., 2023; Yan et al., 2023). For larger dataset, we additionally include a subgraph of ogbn-products and report the evaluation in Appendix I.3.

Table 6: Statistics of our selected datasets.

| Datasets | Nodes | Edges | Domains | Class | Split |
|---|---|---|---|---|---|
| Cornell | 191 | 274 | Acad Webpage | 5 | 48/32/20 |
| Texas | 187 | 280 | Acad Webpage | 5 | 48/32/20 |
| Washington | 229 | 365 | Acad Webpage | 5 | 48/32/20 |
| Wisconsin | 265 | 459 | Acad Webpage | 5 | 48/32/20 |
| arxiv23 | 46,198 | 38,863 | CS Citation | 38 | 60/20/20 |
| Children | 76,875 | 1,162,522 | E-Commerce | 24 | 60/20/20 |
| Amazon | 24,492 | 93,050 | E-Commerce | 5 | 50/25/25 |
| Pubmed | 19,717 | 44,324 | Bio Citation | 3 | 60/20/20 |
| History | 41,551 | 251,590 | E-Commerce | 12 | 60/20/20 |
| Cora | 2,708 | 5,278 | CS Citation | 7 | 60/20/20 |
| citeseer | 3,186 | 4,225 | CS Citation | 6 | 60/20/20 |
| Photo | 48,362 | 436,891 | E-Commerce | 12 | 60/20/20 |
| Computers | 87,229 | 628,274 | E-Commerce | 10 | 72/17/11 |
| Fitness | 173,055 | 1,510,067 | E-Commerce | 13 | 20/10/70 |
| wikics | 11,701 | 215,603 | Knowledge | 10 | 60/20/20 |
| tolokers | 11,758 | 519,000 | Anomaly | 2 | 50/25/25 |

## C.2 CONTENT

**Cornell, Texas, Washington and Wisconsin.** These datasets are collected from web pages of computer science department at various universities. In these datasets, each node represents a web page, while edges are hyperlinks among these pages. In our experiments, we utilize the original webpage data provided by (Yan et al., 2023) as the textual information for each node.

**Cora, citeseer and Pubmed.** These three commonly used citation networks are originally adopted in (Yang et al., 2016), which only provide shallow embeddings with TF-IDF method. Thus, we follow (He et al., 2023; Chen et al., 2024a) to extract the original textual information. Specifically, Cora and citeseer are in the field of computer science, while Pubmed focus on the medical research of diabetes.

**arxiv23.** This dataset is a citation network originally provided by (He et al., 2023), including all cs.ArXiv papers published from January 2023 to September 2023 from the ArXiv daily repository. We adopt their raw texts.

**History, Children, Computers, Photo, Fitness.** These datasets are originally adopted in (Yan et al., 2023), and extracted from (Ni et al., 2019). **(1)** Both of History and Children are extracted from the Amazon-Books dataset, with the second level label "Children" and "History" respectively. The text attribute of node is the title and description of the book itself. **(2)** Both of Computers and Photo are extracted from the Amazon-Electronics dataset, with the second-level label "Computers" and "Photo" respectively. The text attribute of node is the highest voted or randomly selected user review of the item itself. **(3)** Fitness is extracted from the Amazon-Sports dataset, with the second level label "Fitness". The text attribute of node is the title of the item itself. Uniformly, the node in these datasets is the item of different categories, while the edge represents the frequent co-purchased or co-viewed relation.

**wikics.** This dataset is an Internet hyperlink network. Each node is a Wikipedia page with the entry category as label, while each edge is the reference hyperlink. The text attribute of node is the name and content of the entry, which are collected from the official (Mernyei & Cangea, 2020).

**Amazon, tolokers.** These datasets are originally proposed in (Platonov et al., 2023), crawled and transformed into text by (Chen et al., 2024b). **(1)** Amazon is a subgraph based on the Amazon product co-purchasing network metadata from SNAP Datasets (Jure, 2014), where nodes are products (e.g. books, music CDs, DVDs, VHS video tapes), and edges represent frequent co-purchased relations. The text attribute of node is the name of products, while the labels are rating classes. **(2))** tolokers is derived from the Toloka platform (Likhobaba et al., 2023), connecting tolokers (nodes) participated in shared tasks across 13 projects. The text attribute of node is the profile and performance of tolokers, the goal is to predict banned workers in specific projects.

**products.** This dataset is originally provided in (Hu et al., 2020b). We adopt the subgraph extracted by (Chen et al., 2024b) to avoid the high cost of memory, time and finance. Specifically, the node represents an Amazon product with its description as raw text, and the edge between two product indicates the co-purchased relation.

## D DETAILS OF BASELINES

**GCN (Kipf & Welling, 2016), GraphSAGE (Hamilton et al., 2017), GAT (Veličković et al., 2017), and RevGAT (Li et al., 2021)**: These GNNs are commonly used for classification, aggregating information from local neighborhoods to learn node representations.

**GCN-Cheby (Defferrard et al., 2016)**: It uses Chebyshev polynomial approximation for efficient localized spectral filtering, making it particularly suited for large-scale graph data.

**JKNet (Xu et al., 2018)**: It leverages jump knowledge (JK) to combine multi-layer node representations, improving performance on tasks that require both local and global graph structure capture.

**APPNP (Gasteiger et al., 2018)**: It combines personalized PageRank propagation with neural network predictions, enhancing the model's ability to handle complex graph structures.

**H2GCN (Zhu et al., 2020)**: It improves heterophilic graph learning by incorporating higher-order neighbors, separating ego-neighbor embeddings, and utilizing intermediate-layer representations.

**GCNII (Chen et al., 2020)**: It addresses over-smoothing and vanishing gradients through self-supervised signal propagation and exponential moving average mechanisms, enabling stable deep GCN training.

**FAGCN (Bo et al., 2021)**: It utilizes a self-gating mechanism to adaptively integrate low- and high-frequency signals, enabling robust learning on both homophilic and heterophilic graphs.

**GPRGNN (Chien et al., 2020)**: It integrates a generalized, adaptive PageRank propagation mechanism with learnable parameters to dynamically capture both local and global graph structures for improved node representation learning.

**JacobiConv (Wang & Zhang, 2022)**: It eliminates nonlinearity and utilizes Jacobi polynomial bases for spectral filtering, improving flexibility and expressiveness in graph signal learning.

**GBK-GNN (Du et al., 2022)**: It applies a learnable kernel selection mechanism to differentiate homophilic and heterophilic node pairs, optimizing neighborhood aggregation.

**OGNN (Song et al., 2023)**: It introduces an ordered gating mechanism for message passing, enhancing node interactions while mitigating oversmoothing in heterophilic graphs.

**SEGSL (Zou et al., 2023)**: It refines graph topology using structural entropy and encoding trees, improving robustness against noisy edges and adversarial attacks.

**DisamGCL (Zhao et al., 2024)**: It employs topology-aware contrastive learning to disambiguate node embeddings, addressing representation challenges in heterophilic and noisy graphs.

**GNN-SATA (Yang et al., 2024)**: It introduces soft associations between graph topology and node attributes, enabling more effective integration of structural and feature information for graph representation learning.

**TAPE (He et al., 2023)**: It leverages large language models to generate textual explanations, enhancing node classification tasks on text-attributed graphs through an LLM-to-LM interpreter.

**LLM4HeG (Wu et al., 2024)**: It employs a two-stage approach (LLM-enhanced edge discriminator and LLM-guided edge reweighting) for heterophilic graph modeling, with knowledge distillation to transfer capabilities from large to small language models for computational efficiency.

# E COST ESTIMATION

## E.1 TIME ANALYSIS

The time complexity of our proposed LEMP4HG approach is primarily driven by four components: (1) leverage weight-based model $\mathcal{M}^{wb}$ and the paired auxiliary weight-free model $\mathcal{M}^{wf}$ to calculate the heuristic MVRD for edge selection; (2) query LM for connection analysis of selected top-$k$ node pairs with the highest MVRD scores via API calls; (3) encode LM-generated connection analysis from the response into textual embeddings using a fine-tuned SLM; (4) integrate these embeddings into our LM-enhanced message passing mechanism. We further analyze each component as below:

**Heuristic Calculation and Edge Selection** (1) Semi-supervised clustering: the time complexity is $O(n \cdot k \cdot d \cdot \text{iter}) \approx O(n)$ as it typically the case that $n \gg d > k > iter$, where $n$ is the number of nodes, $k$ is the number of clusters, $d$ is the embedding dimension, and iter is the number of iterations. (2) Reliable difference (RD) computation: $O(m \cdot d) \approx O(m)$ for computing pairwise distances across $m$ edges. (3) Variation and modulation (VRD/MVRD) computation: $O(E)$ for simple arithmetic operations per edge; (4) Edge selection: We select top-$k$ edges with the highest MVRD scores from $m$ candidate edges approximately (the initial candidate set includes all the edges, enhanced ones are removed out every $\mathcal{I}$ epochs) by heap-based selection, resulting in a time complexity of $O(m \cdot \log k)$.

**Query LM for Connection Analysis**   This process involves prompt construction, LM inference, response retrieval, and parsing. The overall latency is primarily influenced by the query batch size $k$ and the API rate limits under chat mode, including the maximum query rate $R_q$ (QPM, queries per minute) and maximum token rate $R_t$ (TPM, tokens per minute). For Qwen-turbo in our setup, $R_q = 60$ QPM and $R_t = 1,000,000$ TPM. To mitigate the latency, we employ asynchronous and concurrent processing strategies to improve efficiency. Alternatively, batch mode querying removes rate limits and is better suited for large-scale datasets, though it often exhibits unstable latency.

**Textual Encoding**   The time cost of encoding the LM-generated connection analysis is primarily influenced by the encoder model size and the volume of text in the batch. In our implementation, we use a finetuned DeBERTa-base (He et al., 2021b) with 129 million parameters, which offers a favorable trade-off between efficiency and representation capacity.

**LM-enhanced Message Passing**   The additional computational cost compared to GCN backbone arises in the discriminative message synthesis stage. For each selected node pair, a gating function is applied over the concatenation of node textual embeddings and preliminary messages, involving a matrix-vector multiplication with complexity $O(d^2)$. This leads to a time complexity of $O(\lceil \frac{n_e}{\mathcal{I}} \rceil \cdot k \cdot d^2)$ per layer, where $n_e$ is the current training epoch, $\mathcal{I}$ is the epoch interval for querying LM, $k$ is the batch size for query, and $d$ is the hidden dimension. The message aggregation step retains the standard GCN cost of $O(m \cdot d)$, where $m$ is the number of edges.

In summary, querying LM for connection analysis is the dominant source of runtime overhead in our method. Ignoring other time-consuming components, we can estimate a lower bound on the total runtime in chat mode using the maximum query rate limit (QPM). Specifically, given budget $= \mathcal{B}$ and QPM $= R_q$, the theoretical lower bound on runtime is approximately $\frac{\mathcal{B}}{R_q}$ minutes. If batch mode is adopted, the runtime becomes highly dependent on server-side conditions and is thus difficult to estimate the runtime. Nonetheless, our empirical observations suggest that with a large query batch size (e.g. $\mathcal{B} = 1000$), batch mode typically results in reduced runtime.

To further improve efficiency, several strategies can be considered: (1) deploying a lightweight LM locally; (2) using API services with more relaxed concurrency limits (e.g. Qwen-plus with $R_q = 600$ QPM); (3) or adopting API without explicit concurrency constraints (e.g. Deepseek-v3).

### E.2   MEMORY ANALYSIS

Unlike GCN, which only maintains node-level representations with a space complexity of $O(n \cdot d)$, our method additionally stores edge-level LM-enhanced messages, incurring an extra memory cost of $O(\frac{n_e}{\mathcal{I}} \cdot k \cdot d)$ at training epoch $n_e$. While this design enables more expressive and informative message representations, it also increases overall memory footprint, particularly for dense graphs.

### E.3   FINANCIAL COST ANALYSIS

We adopt Qwen-turbo as LM to generate connection analysis between selected node pairs. According to the pricing scheme by API calls, the model incurs a cost of \$0.02 per million tokens for input (prompt) and \$0.04 per million tokens for output (completion). To estimate the financial cost, we report the average number of input and output tokens per query across 16 datasets in Table 7, using the DeBERTa-base tokenizer. Token consumption varies notably across datasets due to differences in node description length and prompt structure. For example, wikics has the highest average input length (3,195 tokens), resulting in a cost of \$0.76 per 10,000 queries, while lightweight datasets such as Amazon and Fitness require less than \$0.15 for the same number of queries. Despite such variations in input size, the average output length remains relatively stable (around 160–190 tokens).

Table 7: Statistics of token usage and associated costs. "prompt" and "completion" refer to the average token counts per query, and "cost" denotes the estimated cost (USD) for 10,000 queries.

| | Cornell | Texas | Wash. | Wis. | arxiv23 | Child | Amazon | Pubmed | History | Cora | citeseer | Photo | Comp. | Fitness | wikics | tolokers |
|---|---|---|---|---|---|---|---|---|---|---|---|---|---|---|---|---|
| prompt | 1240 | 1024 | 1081 | 1439 | 700 | 672 | 185 | 905 | 667 | 507 | 560 | 712 | 412 | 211 | 3195 | 325 |
| completion | 175 | 176 | 173 | 176 | 176 | 167 | 191 | 171 | 189 | 164 | 163 | 145 | 147 | 182 | 158 | 163 |
| cost ($) | 0.32 | 0.30 | 0.31 | 0.36 | 0.26 | 0.25 | 0.11 | 0.28 | 0.27 | 0.23 | 0.24 | 0.25 | 0.20 | 0.15 | 0.76 | 0.16 |

### E.4 Case Study

To further address the potential concern about the scalability, we report a more explicit analysis of time and financial cost vs. accuracy, using the Pubmed dataset as a representative case. The results are shown in Tables 8 and 9. Specifically, to obtain edge-level connection analysis, we use the Qwen-turbo API in chat mode, with a cost of \$0.04 per million input tokens and \$0.08 per million output tokens. We simulate two realistic usage scenarios:

**One-off scenario (Table 8)**: Each experiment assumes no prior LM-generated connection analysis. All selected edge pairs are queried from scratch under given budget, simulating a cold-start setting.

**Incremental scenario (Table 9)**: Previously queried connection analyses are reused when the dataset grows or the budget increases. Only newly selected edge pairs are queried, reflecting a more practical and cost-efficient usage in dynamic or expanding graphs.

Table 8: Cost-accuracy relation of our LEMP4HG on Pubmed. (One-off Scenario)

| Budget | 0 | 500 | 1000 | 2000 | 3000 | 5000 | 7000 | 10000 |
|---|---|---|---|---|---|---|---|---|
| Financial Cost (\$) | 0 | 0.02 | 0.03 | 0.06 | 0.09 | 0.15 | 0.21 | 0.30 |
| Running Time (s) | 4.84 | 201.04 | 330.11 | 636.93 | 1110.56 | 1326.72 | 1764.09 | 2784.07 |
| Accuracy | 0.9346 | 0.9350 | 0.9381 | 0.9412 | 0.9441 | 0.9457 | 0.9470 | 0.9485 |

Table 9: Cost-accuracy relation of our LEMP4HG on Pubmed. (Incremental Scenario)

| Budget | 0 | 500 | 1000 | 2000 | 3000 | 5000 | 7000 | 10000 |
|---|---|---|---|---|---|---|---|---|
| Financial Cost (\$) | 0 | 0.01 | 0.01 | 0.01 | 0.01 | 0.02 | 0.02 | 0.04 |
| Running Time (s) | 4.45 | 151.28 | 107.85 | 156.04 | 155.92 | 311.23 | 308.23 | 482.03 |
| Accuracy | 0.9346 | 0.9350 | 0.9381 | 0.9412 | 0.9441 | 0.9457 | 0.9470 | 0.9485 |

## F Implementation

### F.1 Experimental Setup

All experiments are conducted on a single NVIDIA A100 GPU with 80GB memory under CentOS 7 with Linux kernel 3.10. The software environment includes PyTorch 2.4.1 with CUDA 12.0 support and PyTorch Geometric 2.6.1, compiled against the PyTorch 2.4 and CUDA 11.8 toolchain.

### F.2 SLM Finetuning

We employ DeBERTa-base (He et al., 2021b) as our SLM for text encoding. The model is finetuned for semi-supervised node classification by appending a one-layer MLP classification head, trained with a cross entropy loss function incorporating label smoothing (0.3). During finetuning, we adopt a training schedule of 4 or 8 epochs, with an initial warm-up phase of 0.6 epochs to stabilize optimization. The learning rate is set to 2e-5, accompanied by a weight decay factor of 0.0 to prevent over-regularization. Dropout regularization is applied with a rate of 0.3 on fully connected layers, and an attention dropout rate of 0.1 is used to mitigate overfitting within the self-attention mechanism. Gradient accumulation steps are set to 1, and training batches consist of 9 samples per device. The parameter settings and training protocol are largely aligned with TAPE (He et al., 2023).

### F.3 LEMP4HG

**GNN Training** We employ a two-layer GCN as the backbone of LEMP4HG, with hidden representations of 128 dimensions. The model is trained for a maximum of 500 epochs using early stopping with a patience of 50 epochs. Optimization is performed using a learning rate of 2e-2, weight decay of 5e-4, and a dropout rate of 0.5. For message synthesis, we adopt the Sigmoid activation function as $\sigma$ in Equation 5, and set the gating coefficient to $\beta = 0.5$ in Equation 6.

**Heuristic Definition** For reliable difference in Equation 8, we set $\gamma = 1.0$ and adopt the Sigmoid function as $\sigma$. For variation in Equation 9, we set $l_a = l_b = 1$. For modulation in Equation 10, we set $\eta = 0.8$ and adopt the Sigmoid function as $\sigma$. Additionally, all distance calculations are batch-normalized, and the final MVRD scores are weighted by $\hat{A}_{\text{sym}}$ to ensure consistency with the message-passing mechanism. The sensitivity analysis of hyper-parameters in the MVRD heuristic is provided in Appendix G.

**Active Leaning for Edge Selection** For time-sensitive weight in Equation 11, we set $\omega = 0.5$, $\phi = 0.5$, $\mathcal{I} = 10$ epochs. Additionally, the high-dimensional input $X$ from SLM's hidden layer (e.g. 768-dimension) may hinder the effectiveness of $\mathcal{M}^{wf}$ without a projection matrix. Thus, we apply PCA (e.g. reducing to 128 dimensions) before feeding $X$ into $\mathcal{M}^{wf}$.

## F.4 SEMI-SUPERVISED CLUSTERING

We define the original node representations as $X$, the neighboring function as $\mathcal{N}_{nei}(\cdot)$. $v_j \in \mathcal{N}_{sel}(v_i)$ when node pair $(v_i, v_j)$ is selected to query for connection analysis, and $m_{ij}$ is the corresponding synthesized message for propagation between $v_i$ and $v_j$. Under one-layer message passing, we define node representations of $v_i$ before and after message passing as $h_i^b$ and $h_i^a$ below:

$$h_i^b = x_i W + b, \quad h_i^a = \sum_{k \in \mathcal{N}_{nei}(v_i) \setminus \mathcal{N}_{sel}(v_i)} \hat{A}_{sym}^{ki} \cdot h_k^b + \sum_{k \in \mathcal{N}_{sel}(v_i)} \hat{A}_{sym}^{ki} \cdot m_{ki} \quad (13)$$

Then we utilize the labeled $h^{b,l}$ and $h^{a,l}$ to calculate the cluster centers $c_k^b$ and $c_k^a$ respectively for each class $k \in \{0, 1, ..., K\}$ and obtain the pseudo labels $\hat{y}_i^b$ and $\hat{y}_i^a$ for all the nodes as below:

$$c_k = \frac{1}{|\{i | \hat{y}_i = k\}|} \sum_{i: \hat{y}_i = k} h_i, \quad \forall k \in \{0, ..., K\} \quad (14)$$

$$\hat{y}_i = \arg \min_k \|h_i - c_k\|^2, \quad h_i \in \{h_i^b, h_i^a\} \quad (15)$$

We re-calculate the cluster centers with pseudo labels $\hat{y}_i$ and all $h_i^b$ and $h_i^a$ as below:

$$c_k = \frac{1}{|\{i | \hat{y}_i = k\}|} \sum_{i: \hat{y}_i = k} h_i, \quad \forall k \in \{0, 1, ..., K\} \quad (16)$$

## G  HYPER-PARAMETER SENSITIVITY STUDY

To evaluate the sensitivity of our heuristic MVRD design, we provide experimental results on two key hyperparameters: $\gamma$ in Equation 8, which controls the scaling of pairwise embedding distances, and $\eta$ in Equation 10, which modulates benign convergence effects. Tables 10–13 report test accuracy on four representative datasets (Pubmed, arxiv23, wikics, and Children) across a grid of $\gamma$ and $\eta$ settings, under specific LM query budgets.

From these results, we can have the following observations:

- Pubmed and arxiv23 (budget = 10,000): Accuracy is highly stable across all $\gamma$ and $\eta$ values, with variation within ± 0.0002, suggesting MVRD is robust in moderately sized scenarios.
- wikics and Children (larger graphs / higher budgets): Accuracy varies slightly more, but peaks consistently around $\gamma = 1.0$ and $\eta \approx 0.8$, indicating these defaults provide strong generalization.

These results suggest that while MVRD is more sensitive in resource-intensive settings, our default configuration ($\gamma = 1.0, \eta = 0.8$) performs robustly across diverse datasets. Other auxiliary parameters (e.g. query interval) relate more to the training process and will be explored in future work.

Table 10: Test accuracy across different $\gamma$ and $\eta$ configurations for Pubmed with budget-10,000

| $\gamma \backslash \eta$ | 0.4 | 0.6 | 0.8 | 1.0 | 1.2 | 1.4 | avg |
|---|---|---|---|---|---|---|---|
| 0.5 | 0.9473 | 0.9463 | 0.9464 | 0.9468 | 0.9470 | 0.9457 | 0.9466 |
| 0.75 | 0.9464 | 0.9466 | 0.9466 | 0.9469 | 0.9470 | 0.9464 | 0.9467 |
| 1.0 | 0.9473 | 0.9471 | 0.9462 | 0.9464 | 0.9466 | 0.9468 | 0.9467 |
| 1.5 | 0.9466 | 0.9466 | 0.9461 | 0.9473 | 0.9462 | 0.9471 | 0.9467 |
| 2.0 | 0.9464 | 0.9471 | 0.9468 | 0.9461 | 0.9471 | 0.9477 | 0.9469 |
| avg | 0.9468 | 0.9467 | 0.9464 | 0.9467 | 0.9468 | 0.9467 | |

Table 11: Test accuracy across different $\gamma$ and $\eta$ configurations for arxiv23 with budget-10,000

| $\gamma \backslash \eta$ | 0.4 | 0.6 | 0.8 | 1.0 | 1.2 | 1.4 | avg |
|---|---|---|---|---|---|---|---|
| 0.5 | 0.7833 | 0.7836 | 0.7843 | 0.7846 | 0.7839 | 0.7833 | 0.7838 |
| 0.75 | 0.7842 | 0.7830 | 0.7841 | 0.7844 | 0.7839 | 0.7836 | 0.7839 |
| 1.0 | 0.7836 | 0.7849 | 0.7843 | 0.7840 | 0.7837 | 0.7834 | 0.7840 |
| 1.5 | 0.7841 | 0.7837 | 0.7837 | 0.7837 | 0.7835 | 0.7850 | 0.7840 |
| 2.0 | 0.7843 | 0.7841 | 0.7830 | 0.7836 | 0.7840 | 0.7835 | 0.7838 |
| avg | 0.7839 | 0.7839 | 0.7839 | 0.7841 | 0.7838 | 0.7838 | |

Table 12: Test accuracy across different $\gamma$ and $\eta$ configurations for wikics with budget-20,000

| $\gamma \backslash \eta$ | 0.4 | 0.6 | 0.8 | 1.0 | 1.2 | 1.4 | avg |
|---|---|---|---|---|---|---|---|
| 0.5 | 0.8732 | 0.8743 | 0.8743 | 0.8727 | 0.8737 | 0.8744 | 0.8738 |
| 0.75 | 0.8745 | 0.8739 | 0.8738 | 0.8749 | 0.8742 | 0.8740 | 0.8742 |
| 1.0 | 0.8725 | 0.8742 | 0.8714 | 0.8749 | 0.8747 | 0.8751 | 0.8738 |
| 1.5 | 0.8745 | 0.8738 | 0.8737 | 0.8743 | 0.8749 | 0.8739 | 0.8742 |
| 2.0 | 0.8728 | 0.8741 | 0.8720 | 0.8742 | 0.8742 | 0.8738 | 0.8735 |
| avg | 0.8735 | 0.8741 | 0.8730 | 0.8742 | 0.8743 | 0.8742 | |

Table 13: Test accuracy across different $\gamma$ and $\eta$ configurations for Children with budget-50,000

| $\gamma \backslash \eta$ | 0.4 | 0.6 | 0.8 | 1.0 | 1.2 | 1.4 | avg |
|---|---|---|---|---|---|---|---|
| 0.5 | 0.6142 | 0.6158 | 0.6166 | 0.6160 | 0.6148 | 0.6144 | 0.6153 |
| 0.75 | 0.6155 | 0.6156 | 0.6171 | 0.6154 | 0.6153 | 0.6151 | 0.6157 |
| 1.0 | 0.6173 | 0.6167 | 0.6166 | 0.6170 | 0.6142 | 0.6166 | 0.6164 |
| 1.5 | 0.6141 | 0.6159 | 0.6129 | 0.6159 | 0.6152 | 0.6163 | 0.6151 |
| 2.0 | 0.6148 | 0.6149 | 0.6150 | 0.6132 | 0.6162 | 0.6158 | 0.6150 |
| avg | 0.6152 | 0.6158 | 0.6156 | 0.6155 | 0.6151 | 0.6156 | |

# H  THEORETICAL ANALYSIS

## H.1  MONOTONICITY OF RELIABLE DIFFERENCE

**Theorem 1** *Let $\gamma > 0$ and $\sigma : \mathbb{R} \to \mathbb{R}^+$ be strictly increasing. Then the reliable difference $RD_{ij}$ is strictly increasing w.r.t. $d_{ij}$, and strictly decreasing w.r.t. $d_i$ and $d_j$.*

**Proof H.1** *By the quotient rule,*

$$\frac{\partial RD_{ij}}{\partial d_{ij}} = \frac{\gamma \, \sigma'(\gamma d_{ij}) \, \sigma(d_i^c + d_j^c)}{\sigma(d_i^c + d_j^c)^2} = \frac{\gamma \, \sigma'(\gamma d_{ij})}{\sigma(d_i^c + d_j^c)} > 0, \tag{17}$$

*since $\sigma' > 0$ and $\sigma(\cdot) > 0$. Similarly,*

$$\frac{\partial RD_{ij}}{\partial d_i^c} = -\frac{\sigma(\gamma d_{ij}) \, \sigma'(d_i^c + d_j^c)}{\sigma(d_i^c + d_j^c)^2} < 0, \tag{18}$$

*and likewise for $d_j^c$. Hence $RD_{ij}$ increases with $d_{ij}$ and decreases with each $d_k^c$.*

# I  DETAILED EXPERIMENTAL EVALUATION

## I.1  MAIN EXPERIMENTS

We conduct four independent experiments for each setting using different random seeds. The Table 23 and 24 present the averaged results along with standard deviations on heterophilic and homophilic graph datasets respectively.

## I.2  COMPARISON WITH LLM4HEG

LLM4HeG is the closest baseline leveraging language models for heterophilic graph learning by discriminating edges to guide reweighting. We conduct experiments to evaluate both methods under consistent settings on multiple datasets, with the performance comparison summarized in Table 14.

The results demonstrate that our LEMP4HG outperforms LLM4HeG on most datasets. Our method achieves consistently better performance on medium-scale benchmarks including Cora, citeseer, and Pubmed. For the smaller datasets (Cornell, Texas, Wisconsin, and Washington), both methods exhibit comparable performance with higher variance due to the limited dataset sizes.

Specifically, we encode node text with DeBERTa-base and using Vicuna-7B for both fine-tuning and edge-level inference, following LLM4HeG's methodology. It is worth noting that LLM4HeG involves computationally expensive processes including LLM fine-tuning, LLM-to-SLM distillation, and complete edge-level inference, which limits its scalability to larger graphs. This computational constraint likely explains why LLM4HeG does not report results on large-scale graph datasets.

The comparative results validate the effectiveness and practical advantages of LEMP4HG over the LLM4HeG approach, particularly in terms of scalability and computational efficiency.

Table 14: Performance comparison between LLM4HeG and our LEMP4HG

| Dataset | Cornell | Texas | Wisconsin | Washington | Cora | citeseer | Pubmed |
|---|---|---|---|---|---|---|---|
| LLM4HeG | 0.8524±0.0213 | **0.8362±0.0181** | **0.8696±0.0160** | 0.8245±0.0276 | 0.8283±0.0027 | 0.7418±0.0020 | 0.9440±0.0016 |
| LEMP4HG (Ours) | **0.8526±0.0922** | 0.8269±0.0245 | 0.8564±0.0106 | **0.8981±0.0576** | **0.8803±0.0145** | **0.7888±0.0078** | **0.9485±0.0042** |

## I.3 Scalability

**Budget**   We present the results of the budget scalability study on the remaining datasets in Tables 6 and 7. For the four small datasets in $f_4$, we enhance all edges directly, without varying the budget $\mathcal{B}$ for scalability study. Taken together with the results in Table 3, we observe a consistent trend that aligns with our proposed budget allocation guidelines in Section 4.4. On the three large homophilic graph datasets—Photo, Computer, and Fitness—which exhibit benign message passing effect, performance tends to degrade as the budget increases, i.e. more enhanced messages integrated. Nonetheless, it remains an open question whether setting the budget $\mathcal{B}$ to a level comparable to the number of edges could potentially reverse this trend and improve the performance, which warrants further investigation. For all other categories of datasets, our suggested budget allocation strategy consistently yields performance improvements. The only exception is Amazon, which exhibits a fluctuating performance curve. This instability can be attributed to the low-quality node textual content—specifically, product names—which limits the LM's ability to generate effective connection analysis and hinders the alignment between message representations and node textual embeddings for significant disparity in text length.

**Large-scale dataset**   The dataset products is a subgraph extracted from ogbn-products, with statistics shown in Table 15. We further categorize it based on structural homophily and message-passing effects, as shown in Table 16. According to the node and edge homophily scores ($H_{node}$ and $H_{edge}$), the dataset falls under the homophily category in H-Cat. Meanwhile, the performance gains from message passing places it in the benign category under MP-Cat.

According to our budget allocation guidelines in the Section 4.4, for large-scale homophilic datasets with benign message-passing effects, we recommend not allocating much budget for message enhancements, as traditional GNNs can achieve satisfactory performance efficiently.

From Table 17, we can observe a steady improvement in performance as the budget increases, which demonstrates the scalability of our proposed LEMP4HG on large graphs.

As shown in Table 18, our LEMP4HG outperforms most general GNNs and several heterophily-specific methods, while maintaining efficient training on a large-scale graph. Although GPRGNN and JacobiConv achieve slightly better accuracy, this result is consistent with our guideline: for large homophilic graphs with benign message passing, lightweight GNNs such as GPRGNN and JacobiConv are already effective, and thus further budget allocation is unnecessary.

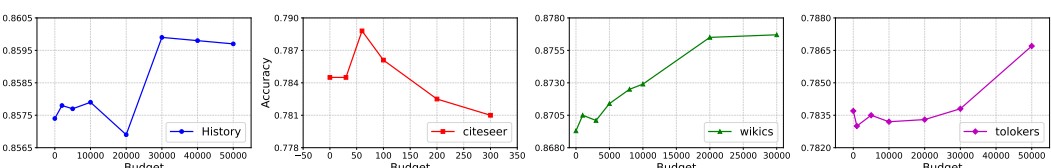

Figure 6: Scalability study on History, citeseer, wikics and tolokers: accuracy v.s. budget

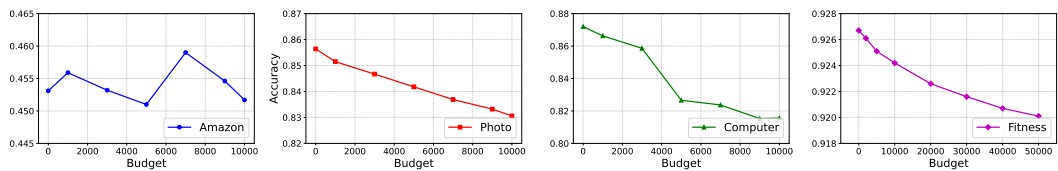

Figure 7: Scalability study on Amazon, Photo, Computer and Fitness: accuracy v.s. budget

Table 15: Statistics of dataset products (subgraph of ogbn-products).

| Datasets | Nodes | Edges | Domains | Class | Split |
|----------|-------|-------|---------|-------|-------|
| products | **316,513** | **9,668,861** | E-Commerce | 39 | 8/2/90 |

Table 16: Categorization of dataset products.

| H-Cat | MP-Cat | Dataset | $H_{node}$ | $H_{edge}$ | 2-MLP | 4-MLP | 2-GCN | 4-GCN |
|-------|--------|---------|-----------|-----------|-------|-------|-------|-------|
| **homophily** | **benign** | products | 0.7971 | 0.8081 | 0.8519 ± 0.0011 | 0.8499 ± 0.0017 | **0.8871 ± 0.0006** | **0.8801 ± 0.0015** |

Table 17: Scalability study on dataset products: accuracy vs. budget.

| Budget | 0 | 10000 | 30000 | 50000 | 70000 | 100000 | 150000 | 200000 |
|--------|---|-------|-------|-------|-------|--------|--------|--------|
| Accuracy | 0.8882 | 0.8887 | 0.8885 | 0.8901 | 0.8905 | 0.8913 | 0.8927 | 0.8939 |

Table 18: Model evaluation on dataset products. The remaining models show OOM or OOT.

| Model | MLP | GCN | SAGE | GAT | RevGAT | Cheby | JKNet | APPNP | GCNII | GPR | Jacobi | OGNN | LEMP4HG |
|-------|-----|-----|------|-----|--------|-------|-------|-------|-------|-----|--------|------|---------|
| Accuracy | 0.8634 | 0.8868 | 0.8933 | 0.8829 | 0.8859 | 0.8634 | 0.8862 | 0.8634 | 0.8738 | **0.8953** | **0.8968** | 0.8908 | **0.8939** |

## J  PROMPT DESIGN

### J.1  PROMPT FOR CONNECTION ANALYSIS

We query LM for connection analysis between paired nodes by providing their associated textual content. These texts typically contain basic information about the entities, such as the title and abstract of a paper, or the description and user reviews of a product. Detailed descriptions of the node textual content of each dataset are provided in Appendix C.2. The prompt templates designed for each dataset are presented in Tables 19 and 20. Note that, Acad Webpage dataset includes Cornell, Texas, Washington and Wisconsin, while CS Citation dataset includes Cora, citeseer and arxiv23. These two categories of datasets share the same prompt template respectively.

### J.2  PROMPT FOR PREDICTION AND EXPLANATION

As TAPE (He et al., 2023), we query LM for category prediction and explanation to enhance node representations. While TAPE originally refers to using the title and abstract as input for citation networks, we adopt a more generalized interpretation in our study. Specifically, the input to the LM is defined as the basic information of each entity, which varies across datasets—for instance, it includes the title and abstract in citation networks, and the product description in e-commerce networks. The corresponding prompt templates are detailed in Tables 21 and 22.

### J.3  TEXTUAL GENERATION

we present a sample connection analysis generated for a node pair in the Pubmed dataset as below.

*The relational implications between Paper A and Paper B are as below. Paper A investigates the association between type 2 diabetes and osteopenia, finding variable BMD outcomes depending on age and gender. It emphasizes the need for individualized assessment. Paper B reviews the role of insulin as an anabolic agent in bone, suggesting that insulin may influence bone density and strength in diabetic patients. The intellectual connection lies in their shared focus on diabetes and bone health, with Paper A providing empirical data and Paper B offering a mechanistic explanation involving insulin's role. Together, they highlight the complex relationship between diabetes and bone metabolism, with Paper B potentially explaining the findings observed in Paper A.*

## K  LLM USAGE STATEMENT

Large language models were used solely for language polishing and improving readability during manuscript preparation, including sentence rephrasing and grammar checks. The authors take full responsibility for all research ideas, methodologies, experiments, and scientific content. The LLM played no role in conceptualization, analysis, or experimental design.

Table 19: Prompt templates for querying connection analysis by LM: Part 1

| Dataset | Prompt |
| --- | --- |
| Acad Webpage | Analyze the hyperlink relationship between Webpage A and Webpage B of computer science department of the university, based on their contents provided below. \n \n Your response should: \n 1. Summarize the key content of both webpages and any notable features. \n 2. Clearly explain the intellectual connection or relevance between the two webpages, highlighting how they might be related. \n 3. Keep the response concise (within 200 words) and ensure it emphasizes the connection between the two webpages. \n 2 4. Use the following sentence structure: "The relational implications between [Webpage A] and [Webpage B] are as below." ence department of the university, based on their contents provided below. \n \n Webpage A: <content A>. Webpage B: <content B> |
| CS Citation | Analyze the citation relationship between Paper A and Paper B in the filed of computer science, based on their titles and abstracts provided below. \n \n Your response should: \n 1. Summarize the key content of both papers, focusing on their main research questions, methods, findings, and contributions. \n 2. Clearly explain the intellectual connection or relevance between the two papers, highlighting how they might be related. \n 3. Keep the response concise (within 200 words) and ensure it emphasizes the scholarly connection between the two papers. \n 4. Use the following sentence structure: "The relational implications between [Paper A] and [Paper B] are as below." \n \n Paper A: <content A>. Paper B: <content B> |
| Pubmed | Analyze the citation relationship between Paper A and Paper B in the filed of medical research of diabetes, based on their titles and abstracts provided below. \n \n Your response should: \n 1. Summarize the key content of both papers, focusing on their main research questions, methods, findings, and contributions. \n 2. Clearly explain the intellectual connection or relevance between the two papers, highlighting how they might be related. \n 3. Keep the response concise (within 200 words) and ensure it emphasizes the scholarly connection between the two papers. \n 4. Use the following sentence structure: "The relational implications between [Paper A] and [Paper B] are as below." \n \n Paper A: <content A>. Paper B: <content B> |
| History&Children | Analyze the co-purchased or co-viewed relationship between two History / Children-related books sold in the Amazon based on their descriptions and titles provided below. \n \n Your response should: \n 1. Summarize the main points of both items' descriptions. \n 2. Clearly explain the intellectual connection or relevance between the two books, highlighting how they might be related. \n 3. Keep the response concise (within 200 words) and ensure it emphasizes the relationship between the two books. \n 4. Use the following sentence structure: "The relational implications between [Book A] and [Book B] are as below." \n \n Book A: <content A>. Book B: <content B> |
| Photo&Computers | Analyze the co-purchased or co-viewed relationship between two Photo / Computers-related items sold in the Amazon based on their user reviews provided below. \n \n Your response should: \n 1. Summarize the main points of both items' user reviews. \n 2. Clearly explain the intellectual connection or relevance between the two items, highlighting how they might be related. \n 3. Keep the response concise (within 200 words) and ensure it emphasizes the relationship between the two items. \n 4. Use the following sentence structure: "The relational implications between [Item A] and [Item B] are as below." \n \n Item A: <content A>. Item B: <content B> |
| wikics | Analyze the hyperlink relationship between two wikipedia entries based on their titles and contents provided below. \n \n Your response should: \n 1. Summarize the main points of both entries' contents. \n 2. Clearly explain the intellectual connection or relevance between the two entries, highlighting how they might be related. \n 3. Keep the response concise (within 200 words) and ensure it emphasizes the relationship between the two entries. \n 4. Use the following sentence structure: "The relational implications between [Entry A] and [Entry B] are as below." \n \n Entry A: <content A>. Entry B: <content B> |

Table 20: Prompt templates for querying connection analysis by LM: Part 2

| Dataset | Prompt |
|---|---|
| tolokers | Analyze the co-work relationship between two tolokers(workers) based on their profile information and task performance statistics provided below. \n \n Your response should: \n 1. Summarize the main points of both workers' profile and performance. \n 2. Clearly explain the intellectual connection or relevance between the two workers, highlighting how they might be related. \n 3. Keep the response concise (within 200 words) and ensure it emphasizes the relationship between the two toloker. \n 4. Use the following sentence structure: "The relational implications between [Toloker A] and [Toloker B] are as below." \n \n Toloker A: <content A>. Toloker B: <content B> |
| Amazon | Analyze the relationship between two items sold in the Amazon based on their item name. Both items are product like books, music CDs, DVDs, VHS video tapes. \n \n Your response should: \n 1. Describe and summarize the main points of both item. \n 2. Clearly explain the co-purchased or co-viewed relationship (connection or relevance) between the two items. \n 3. Keep the response concise (within 200 words) and ensure it emphasizes the relationship between the two items. \n 4. Use the following sentence structure: "The relational implications between [Item A] and [Item B] are as below." \n \n Item A: <content A>. Item B: <content B> |
| Fitness | Analyze the co-purchased or co-viewed relationship between two Fitness-related items sold in the Amazon based on their item titles provided below. \n \n Your response should: \n 1. Describe and summarize the main points of both item. \n 2. Clearly explain the intellectual connection or relevance between the two items, highlighting how they might be related. \n 3. Keep the response concise (within 200 words) and ensure it emphasizes the relationship between the two items. \n 4. Use the following sentence structure: "The relational implications between [Item A] and [Item B] are as below." \n \n Item A: <content A>. Item B: <content B> |
| products | Analyze the co-purchased relationship between two items sold in the Amazon based on their product descriptions provided below. \n \n Your response should: \n 1. Summarize the main points of both products' descriptions. \n 2. Clearly explain the intellectual connection or relevance between the two items, highlighting how they might be related. \n 3. Keep the response concise (within 200 words) and ensure it emphasizes the relationship between the two items. \n 4. Use the following sentence structure: "The relational implications between [Item A] and [Item B] are as below." \n \n Item A: <content A>. Item B: <content B> |

Table 21: Prompt templates for querying category prediction and explanation by LM: Part 1

| Dataset | Prompt |
|---|---|
| Acad Webpage | [Webpage Content]: <content> \n \n [Question]: Which of the following categories does this webpage belong to: Student, Project, Course, Staff, Faculty? If multiple options apply, provide a comma-separated list ordered from most to least related, then for each choice you gave, explain how it is present in the text. \n \n [Answer]: |
| Pubmed | [Paper Info]. <content> \n \n [Question]: Does the paper involve any cases of Type 1 diabetes, Type 2 diabetes, or Experimentally induced diabetes? Please give one or more answers of either Type 1 diabetes, Type 2 diabetes, or Experimentally induced diabetes; if multiple options apply, provide a comma-separated list ordered from most to least related, then for each choice you gave, give a detailed explanation with quotes from the text explaining why it is related to the chosen option. \n \n [Answer]: |
| arxiv23 | [Paper Info]. <content> \n \n [Question]: Which arXiv CS subcategory does this paper belong to? Give 5 likely arXiv CS sub-categories as a comma-separated list ordered from most to least likely, in the form "cs.XX", and provide your reasoning. \n \n [Answer]: |
| History | [Book Info]. <content> \n \n [Question]: Which of the following sub-categories of History does this book belong to: World, Americas, Asia, Military, Europe, Russia, Africa, Ancient Civilizations, Middle East, Historical Study & Educational Resources, Australia & Oceania and Arctic & Antarctica? If multiple options apply, provide a comma-separated list ordered from most to least related, then for each choice you gave, explain how it is present in the text. \n \n [Answer]: |

Table 22: Prompt templates for querying category prediction and explanation by LM: Part 2

| Dataset | Prompt |
|---------|--------|
| Children | [Book Info]. \<content\> \n \n [Question]: Which of the following sub-categories of Children does this book belong to: Literature & Fiction, Animals, Growing Up & Facts of Life, Humor, Cars Trains & Things That Go, Fairy Tales Folk Tales & Myths, Activities Crafts & Games, Science Fiction & Fantasy, Classics, Mysteries & Detectives, Action & Adventure, Geography & Cultures, Education & Reference, Arts Music & Photography, Holidays & Celebrations, Science Nature & How It Works, Early Learning, Biographies, History, Children's Cookbooks, Religions, Sports & Outdoors, Comics & Graphic Novels, Computers & Technology? If multiple options apply, provide a comma-separated list ordered from most to least related, then for each choice you gave, explain how it is present in the text. \n \n [Answer]: |
| Cora | [Paper Info]. \<content\> \n \n [Question]: Which of the following sub-categories of AI does this paper belong to: Case Based, Genetic Algorithms, Neural Networks, Probabilistic Methods, Reinforcement Learning, Rule Learning, Theory? If multiple options apply, provide a comma-separated list ordered from most to least related, then for each choice you gave, explain how it is present in the text. \n \n [Answer]: |
| citeseer | [Paper Info]. \<content\> \n \n [Question]: Which of the following sub-categories of computer science does this paper belong to: Agents, Machine Learning, Information Retrieval, Database, Human-Computer Interaction and Artificial Intelligence? If multiple options apply, provide a comma-separated list ordered from most to least related, then for each choice you gave, explain how it is present in the text. \n \n [Answer]: |
| wikics | [Entry info]: \<content\> \n \n [Question]: Which of the following sub-categories of computer science does this wikipedia entry belong to: Computational linguistics, Databases, Operating systems, Computer architecture, Computer security, Internet protocols, Computer file systems, Distributed computing architecture, Web technology and Programming language topics? If multiple options apply, provide a comma-separated list ordered from most to least related, then for each choice you gave, explain how it is present in the text. \n \n [Answer]: |
| tolokers | [Worker info]. \<content\> \n \n [Question]: What is the probability that the worker will be banned from a specific project? Choose one of the following options: Very Low (¡10%), Low (10-30%), Moderate (30-50%), High (50-70%), Very High (¿70%). Then, explain how the choice you gave is present in the text. \n \n [Answer]: |
| Amazon | [Item name]. \<content\> \n \n [Question]: It is a product like books, music CDs, DVDs, VHS video tapes. What is the grade that the item will be rated? Choose one of the following options: Good (score 5-3.5), Average (score 3.5-2.5), Bad (score 2.5-1). Then, explain how the choice you gave is present in the text. \n \n [Answer]: |
| Photo | [Item review]. \<content\> \n \n [Question]: Which of the following sub-categories of photo does this electronic item belong to: Film Photography, Video, Digital Cameras, Accessories, Binoculars & Scopes, Lenses, Bags & Cases, Lighting & Studio, Flashes, Tripods & Monopods, Underwater Photography, Video Surveillance? If multiple options apply, provide a comma-separated list ordered from most to least related, then for each choice you gave, explain how it is present in the text. \n \n [Answer]: |
| Computers | [Item review]. \<content\> \n \n [Question]: Which of the following sub-categories of computer does this electronic item belong to: Laptop Accessories, Computer Accessories & Peripherals, Computer Components, Data Storage, Networking Products, Monitors, Computers & Tablets, Tablet Accessories, Servers, Tablet Replacement Parts? If multiple options apply, provide a comma-separated list ordered from most to least related, then for each choice you gave, explain how it is present in the text. \n \n [Answer]: |
| Fitness | [Item title]. \<content\> \n \n [Question]: Which of the following sub-categories of fitness does this item belong to: Other Sports, Exercise & Fitness, Hunting & Fishing, Accessories, Leisure Sports & Game Room, Team Sports, Boating & Sailing, Swimming, Tennis & Racquet Sports, Golf, Airsoft & Paintball, Clothing, Sports Medicine? If multiple options apply, provide a comma-separated list ordered from most to least related, then for each choice you gave, explain how it is present in the text. \n \n [Answer]: |

Table 23: Detailed evaluation on heterophilic graph datasets.

| Datasets | Cornell | Texas | Washington | Wisconsin | arxiv23 | Children | Amazon |
|---|---|---|---|---|---|---|---|
| MLP | 0.8654 ± 0.0674 | 0.8462 ± 0.0000 | 0.8404 ± 0.0662 | 0.8796 ± 0.0685 | 0.7811 ± 0.0035 | 0.6199 ± 0.0071 | 0.4275 ± 0.0087 |
| GCN | 0.6346 ± 0.0606 | 0.6026 ± 0.0797 | 0.6596 ± 0.0174 | 0.5972 ± 0.1073 | 0.7785 ± 0.0023 | 0.6083 ± 0.0061 | 0.4558 ± 0.0140 |
| SAGE | 0.8269 ± 0.0922 | 0.8269 ± 0.0128 | 0.8564 ± 0.0319 | 0.8935 ± 0.0699 | 0.7861 ± 0.0028 | 0.6245 ± 0.0055 | 0.4648 ± 0.0352 |
| GAT | 0.4808 ± 0.0922 | 0.5962 ± 0.1075 | 0.5532 ± 0.0796 | 0.4769 ± 0.0847 | 0.7622 ± 0.0061 | 0.5824 ± 0.0057 | 0.4520 ± 0.0203 |
| RevGAT | 0.8397 ± 0.0674 | 0.8205 ± 0.0000 | 0.8777 ± 0.0403 | 0.8935 ± 0.0762 | 0.7798 ± 0.0046 | 0.6195 ± 0.0054 | 0.4590 ± 0.0184 |
| GCN-Cheby | 0.8654 ± 0.0674 | 0.8462 ± 0.0000 | 0.8404 ± 0.0662 | 0.8796 ± 0.0685 | 0.7811 ± 0.0035 | 0.6199 ± 0.0071 | 0.4275 ± 0.0087 |
| JKNet | 0.6603 ± 0.0766 | 0.6410 ± 0.0363 | 0.7181 ± 0.0319 | 0.6667 ± 0.0741 | 0.7774 ± 0.0033 | 0.6031 ± 0.0082 | 0.4551 ± 0.0116 |
| APPNP | 0.6474 ± 0.0847 | 0.6538 ± 0.0610 | 0.7500 ± 0.0363 | 0.6250 ± 0.0762 | 0.7762 ± 0.0021 | 0.6241 ± 0.0080 | 0.4534 ± 0.0093 |
| H2GCN | 0.6795 ± 0.1936 | 0.7244 ± 0.0736 | 0.8138 ± 0.0822 | 0.7639 ± 0.1652 | 0.7761 ± 0.0044 | 0.6126 ± 0.0039 | 0.4071 ± 0.0323 |
| GCNII | 0.8013 ± 0.0990 | 0.8013 ± 0.0641 | 0.8564 ± 0.0472 | 0.8750 ± 0.0699 | 0.7832 ± 0.0036 | 0.6223 ± 0.0059 | 0.4429 ± 0.0230 |
| FAGCN | 0.7051 ± 0.2308 | 0.7949 ± 0.0209 | 0.7074 ± 0.1914 | 0.8102 ± 0.1268 | 0.7446 ± 0.0707 | 0.6287 ± 0.0029 | 0.4319 ± 0.0301 |
| GPRGNN | 0.8269 ± 0.0641 | 0.8333 ± 0.0256 | 0.8564 ± 0.0635 | 0.8796 ± 0.0778 | 0.7815 ± 0.0022 | 0.6316 ± 0.0076 | 0.4554 ± 0.0128 |
| JacobiConv | 0.7756 ± 0.1180 | 0.7692 ± 0.0573 | 0.7766 ± 0.0238 | 0.8426 ± 0.0711 | 0.7153 ± 0.0076 | 0.5981 ± 0.0223 | 0.4554 ± 0.0086 |
| GBK-GNN | 0.8333 ± 0.0529 | 0.8397 ± 0.0213 | 0.8085 ± 0.0620 | 0.8889 ± 0.0600 | 0.7617 ± 0.0069 | 0.4961 ± 0.0199 | 0.4274 ± 0.0058 |
| OGNN | 0.8462 ± 0.0654 | 0.8397 ± 0.0213 | 0.8564 ± 0.0409 | 0.8981 ± 0.0499 | 0.7820 ± 0.0030 | 0.6250 ± 0.0048 | 0.4366 ± 0.0109 |
| SEGSL | 0.8333 ± 0.0529 | 0.8590 ± 0.0529 | 0.8564 ± 0.0485 | 0.9028 ± 0.0479 | OOT | OOT | OOT |
| DisamGCL | 0.8462 ± 0.0831 | 0.8141 ± 0.0525 | 0.8404 ± 0.0681 | 0.8704 ± 0.0655 | 0.7801 ± 0.0037 | OOM | 0.4410 ± 0.0115 |
| GNN-SATA | 0.8141 ± 0.0838 | 0.8077 ± 0.0385 | 0.8457 ± 0.0570 | 0.8935 ± 0.0660 | OOM | OOM | 0.4237 ± 0.0088 |
| TAPE+SAGE | 0.8718 ± 0.0468 | 0.8526 ± 0.0641 | 0.8670 ± 0.0745 | 0.8889 ± 0.0338 | 0.8023 ± 0.0028 | 0.6310 ± 0.0061 | 0.4639 ± 0.0237 |
| TAPE+RevGAT | 0.8846 ± 0.0534 | 0.8590 ± 0.0331 | 0.8777 ± 0.0703 | 0.9074 ± 0.0262 | 0.7995 ± 0.0056 | 0.6285 ± 0.0060 | 0.4722 ± 0.0103 |
| LEMP | 0.8526 ± 0.0922 | 0.8269 ± 0.0245 | 0.8564 ± 0.0106 | 0.8981 ± 0.0576 | 0.7853 ± 0.0026 | 0.6160 ± 0.0062 | 0.4590 ± 0.0166 |
| LEMP+TAPE | 0.8654 ± 0.0323 | 0.8590 ± 0.0534 | 0.8777 ± 0.0472 | 0.8565 ± 0.0463 | 0.8003 ± 0.0021 | 0.6179 ± 0.0070 | 0.4675 ± 0.0211 |

Table 24: Detailed evaluation on homophilic graph datasets.

| Model | Pubmed | History | Cora | Citeseer | Photo | Computers | Fitness | wikics | tolokers |
|---|---|---|---|---|---|---|---|---|---|
| MLP | 0.9471 ± 0.0043 | 0.8616 ± 0.0052 | 0.8034 ± 0.0161 | 0.7371 ± 0.0116 | 0.7121 ± 0.0013 | 0.6065 ± 0.0044 | 0.8969 ± 0.0010 | 0.8597 ± 0.0060 | 0.7793 ± 0.0096 |
| GCN | 0.9354 ± 0.0021 | 0.8559 ± 0.0053 | 0.8762 ± 0.0166 | 0.7853 ± 0.0128 | 0.8563 ± 0.0012 | 0.8735 ± 0.0019 | 0.9282 ± 0.0004 | 0.8700 ± 0.0033 | 0.7820 ± 0.0029 |
| SAGE | 0.9475 ± 0.0042 | 0.8649 ± 0.0045 | 0.8531 ± 0.0121 | 0.7813 ± 0.0231 | 0.8518 ± 0.0014 | 0.8727 ± 0.0015 | 0.9240 ± 0.0006 | 0.8771 ± 0.0050 | 0.7885 ± 0.0048 |
| GAT | 0.8875 ± 0.0072 | 0.8441 ± 0.0056 | 0.8725 ± 0.0119 | 0.7841 ± 0.0103 | 0.8545 ± 0.0020 | 0.8738 ± 0.0019 | 0.9261 ± 0.0008 | 0.8533 ± 0.0085 | 0.7821 ± 0.0034 |
| RevGAT | 0.9484 ± 0.0028 | 0.8645 ± 0.0054 | 0.8085 ± 0.0235 | 0.7551 ± 0.0148 | 0.7839 ± 0.0009 | 0.7597 ± 0.0010 | 0.9083 ± 0.0007 | 0.8665 ± 0.0069 | 0.7968 ± 0.0087 |
| GCN-Cheby | 0.9471 ± 0.0043 | 0.8616 ± 0.0052 | 0.8034 ± 0.0161 | 0.7371 ± 0.0116 | 0.7114 ± 0.0010 | 0.6045 ± 0.0033 | 0.8969 ± 0.0010 | 0.8597 ± 0.0060 | 0.7793 ± 0.0096 |
| JKNet | 0.9314 ± 0.0069 | 0.8537 ± 0.0038 | 0.8821 ± 0.0163 | 0.7845 ± 0.0104 | 0.8545 ± 0.0023 | 0.8739 ± 0.0018 | 0.9282 ± 0.0006 | 0.8629 ± 0.0056 | 0.7838 ± 0.0029 |
| APPNP | 0.9066 ± 0.0023 | 0.8569 ± 0.0057 | 0.8821 ± 0.0140 | 0.7927 ± 0.0128 | 0.8446 ± 0.0016 | 0.8647 ± 0.0015 | 0.9279 ± 0.0014 | 0.8754 ± 0.0042 | 0.7809 ± 0.0045 |
| H2GCN | 0.9473 ± 0.0045 | 0.8383 ± 0.0051 | 0.8324 ± 0.0534 | 0.7712 ± 0.0210 | 0.8441 ± 0.0014 | 0.8632 ± 0.0019 | 0.9178 ± 0.0010 | 0.8660 ± 0.0057 | 0.7815 ± 0.0037 |
| GCNII | 0.9483 ± 0.0038 | 0.8630 ± 0.0046 | 0.8352 ± 0.0165 | 0.7441 ± 0.0125 | 0.8493 ± 0.0029 | 0.8736 ± 0.0005 | 0.9137 ± 0.0036 | 0.8674 ± 0.0048 | 0.7861 ± 0.0053 |
| FAGCN | 0.8859 ± 0.0845 | 0.7784 ± 0.0951 | 0.8191 ± 0.0728 | 0.7555 ± 0.0285 | 0.8080 ± 0.0292 | 0.7216 ± 0.0902 | 0.7790 ± 0.0822 | 0.8655 ± 0.0103 | 0.7812 ± 0.0082 |
| GPRGNN | 0.9470 ± 0.0029 | 0.8583 ± 0.0059 | 0.8794 ± 0.0146 | 0.7861 ± 0.0182 | 0.8467 ± 0.0055 | 0.8728 ± 0.0006 | 0.9277 ± 0.0017 | 0.8755 ± 0.0039 | 0.7813 ± 0.0041 |
| JacobiConv | 0.9473 ± 0.0027 | 0.8543 ± 0.0057 | 0.8734 ± 0.0122 | 0.7810 ± 0.0152 | 0.8432 ± 0.0007 | 0.8610 ± 0.0042 | 0.9238 ± 0.0020 | 0.8778 ± 0.0048 | 0.7817 ± 0.0030 |
| GBK-GNN | 0.9476 ± 0.0041 | 0.8403 ± 0.0051 | 0.8250 ± 0.0138 | 0.7649 ± 0.0192 | 0.7659 ± 0.0007 | 0.6954 ± 0.0097 | 0.8771 ± 0.0159 | 0.8723 ± 0.0067 | 0.7799 ± 0.0081 |
| OGNN | 0.9487 ± 0.0051 | 0.8633 ± 0.0041 | 0.8066 ± 0.0182 | 0.7504 ± 0.0174 | 0.7914 ± 0.0018 | 0.7693 ± 0.0014 | 0.9095 ± 0.0009 | 0.8684 ± 0.0041 | 0.7803 ± 0.0091 |
| SEGSL | OOT | OOT | 0.8191 ± 0.0201 | 0.7680 ± 0.0101 | OOT | OOT | OOT | OOT | OOT |
| DisamGCL | 0.9476 ± 0.0040 | 0.8604 ± 0.0041 | 0.8103 ± 0.0182 | 0.7343 ± 0.0032 | OOM | OOM | OOM | 0.8651 ± 0.0055 | 0.7835 ± 0.0057 |
| GNN-SATA | 0.9453 ± 0.0037 | OOM | 0.8043 ± 0.0185 | 0.7339 ± 0.0196 | OOM | OOM | OOM | 0.8602 ± 0.0062 | 0.7815 ± 0.0032 |
| TAPE+SAGE | 0.9480 ± 0.0037 | 0.8677 ± 0.0072 | 0.8771 ± 0.0145 | 0.7837 ± 0.0182 | 0.8587 ± 0.0030 | 0.8733 ± 0.0005 | 0.9315 ± 0.0011 | 0.8823 ± 0.0080 | 0.7848 ± 0.0023 |
| TAPE+RevGAT | 0.9480 ± 0.0040 | 0.8664 ± 0.0037 | 0.8439 ± 0.0079 | 0.7774 ± 0.0199 | 0.8002 ± 0.0025 | 0.7640 ± 0.0025 | 0.9215 ± 0.0004 | 0.8824 ± 0.0086 | 0.7991 ± 0.0029 |
| LEMP | 0.9485 ± 0.0042 | 0.8599 ± 0.0036 | 0.8803 ± 0.0145 | 0.7888 ± 0.0078 | Down | Down | Down | 0.8768 ± 0.0052 | 0.7867 ± 0.0025 |
| LEMP+TAPE | 0.9484 ± 0.0029 | 0.8662 ± 0.0064 | 0.8826 ± 0.0116 | 0.7943 ± 0.0143 | 0.8591 ± 0.0028 | 0.8729 ± 0.0021 | 0.9303 ± 0.0008 | 0.8825 ± 0.0040 | 0.7897 ± 0.0054 |

