# OpenReview forum: "Language Model-Enhanced Message Passing for Heterophilic Graph Learning"
_ICLR.cc/2026/Conference — ICLR 2026 Conference Withdrawn Submission_

### Official Review · Reviewer_vfdY · 2025-10-24

**Soundness:** 3
**Presentation:** 2
**Contribution:** 2
**Rating:** 4
**Confidence:** 4

**Summary:**

This paper proposes a novel language model (LM)-enhanced message passing approach for heterophilic graph learning (LEMP4HG).
The method integrates semantic knowledge from paired node texts into the message-passing process by querying a language model to generate connection analyses between node pairs, encoding and fusing the resulting embeddings with node representations through a gating mechanism, and selectively enhancing node pairs that suffer most from message passing via a heuristic
MVRD.

**Strengths:**

1. The paper addresses a relevant and emerging problem, heterophilic graph learning enhanced by language models, which is both practically valuable and interesting.

2. The authors provide a concise summary of existing approaches and their shortcomings, offering a good motivation for the proposed framework.

3. The manuscript is well organized and clearly written.

**Weaknesses:**

1. The proposed approach relies on textual node attributes. However, the abstract and introduction imply a more general scope than the method actually supports.
2. The model's performance is not validated on very large-scale graphs, such as ogbn-arxiv and obgn-products.
3. The model performs worse than RevGAT+T on heterophilic graphs.
4. Some cited works contain outdated or incorrect publication information. For instance,
“Yuxia Wu, Shujie Li, Yuan Fang, and Chuan Shi. Exploring the potential of large language models for heterophilic graphs. arXiv preprint arXiv:2408.14134, 2024.” has already been published in Proceedings of the 2025 Conference of the North American Chapter of the Association for Computational Linguistics: Human Language Technologies (NAACL 2025), pp. 5198–5211.

**Questions:**

1. How does the proposed model perform on non-textual graphs？

---

### Official Review · Reviewer_QSDn · 2025-10-28

**Soundness:** 2
**Presentation:** 2
**Contribution:** 2
**Rating:** 2
**Confidence:** 4

**Summary:**

The paper studies node classification on text-attributed graphs, especially in heterophilic settings where connected nodes have dissimilar features/labels and standard GNN message passing tends to propagate harmful signals. The authors propose LEMP4HG, which augments message passing with language model–generated “connection analysis” for node pairs.
They evaluate on 16 text-attributed graph datasets. They argue prior heterophily work either changes graph structure or modifies aggregation, but usually ignores rich node text

**Strengths:**

1. Instead of just rewiring or redesigning aggregation, LEMP tries to rewrite each edge message using LM-derived “connection analysis,” then fuses that with source/target node embeddings via a discriminative gate. This is a clear, modular mechanism and can plug into a simple GCN backbone.
2. The MVRD heuristic estimates where message passing causes “representation distortion” and queries the LM only on those edges, rather than all edges, to keep LM calls under a budget.

**Weaknesses:**

1. The paper's primary claim is that LEMP4HG "excels on heterophilic graphs". However, the main results in Table 2  contradict this. The proposed LEMP method underperforms the simple 2-layer MLP baseline on several heterophilic datasets, including Cornell (0.8526 vs 0.8654 for MLP), Texas (0.8269 vs 0.8462 for MLP), and Children (0.6160 vs 0.6199 for MLP). On other heterophilic datasets like arxiv23, the gain is marginal (0.7853 vs 0.7811). This undermines the central contribution. This matters because most baselines are lightweight GNNs with standard node features. LEMP4HG uses a pretrained LM (Qwen-turbo) and a finetuned DeBERTa-base SLM. This is a lot more supervision, prior knowledge, and compute costs.
2. The method introduces significant computational and resource costs (API calls to an LM, text encoding with a fine-tuned SLM) compared to pure GNN baselines. The paper's cost analysis for Pubmed (Table 8)  shows that LEMP requires at most 2700 seconds versus the GCN backbone's (budgt 0) ~5 seconds. For a modest 1.48% relative gain in accuracy (0.9485 vs 0.9346), this massive cost is not justified.
3. The paper's contributions are further weakened when compared to other LM-enhanced methods. As shown in Table 2, LEMP is outperformed in average rank by TAPE-enhanced SAGE and RevGAT.
4. The paper identifies LLM4HeG as the only other work that leverages LMs semantically for heterophily, making it the most relevant baseline. However, the direct comparison is relegated to Appendix I.2, obscuring a clear comparison of novelty and performance.
5. In Table 2, the paper reports "Down" for three datasets (Photo, Comp., Fitness) where the method had a "negative impact". This lacks transparency; the concrete negative results should be reported.
6. The paper even reports a “LEMP+T” variant and shows it as among the top performers, which blurs attribution: are the gains from your new “discriminative message synthesis,” or mostly from prior TAPE-style LM augmentation?

**Questions:**

1. Regarding Table 2 , what were the exact performance numbers for the datasets marked as "Down"? Please provide the quantitative results for this "negative impact."
2. The paper claims the "Discriminative Message Synthesis" (Eq. 5-6) addresses the "static nature" of preliminary messages and "misalignment"  with node embeddings. Can you provide a more detailed intuition or theoretical justification for why this specific gating and fusion mechanism is the appropriate solution for these two distinct problems?
3. What is the end-to-end training/inference cost (time, memory, and LM API calls per epoch) versus a vanilla GCN or versus TAPE? Please quantify using the Qwen-turbo setup you describe.
4. What is the precise conceptual and empirical difference between LEMP and LLM4HeG? Why is LLM4HeG only in the appendix when you yourself describe it as the only prior work that “leverages LM to unlock deeper insight under heterophily semantically”?

---

### Official Review · Reviewer_5NNf · 2025-10-28

**Soundness:** 3
**Presentation:** 3
**Contribution:** 3
**Rating:** 6
**Confidence:** 4

**Summary:**

This paper introduces a novel graph learning framework to tackle heterophily in node classification. It leverages LLMs to generate semantic annotations for heterophilous edges (connecting dissimilar nodes), which are fused into node representations to mitigate misleading structural signals. To alleviate LLM inference costs, active learning strategically selects high-uncertainty edges for annotation. Extensive experiments on 16 heterophilous datasets are demonstrate  performance gains over state-of-the-art heterophily-aware GNNs and LLM-augmented baselines.

The paper proposes a graph learning framework for node classification under heterophily. It uses LLMs to generate semantic annotations for edges connecting dissimilar nodes, integrating these annotations into node representations to counter misleading structural signals. To manage computational cost, active learning selects edges for LLM annotation based on uncertainty. The approach is evaluated on 16 datasets for node classification, with comparisons to heterophily-aware GNNs and LLM-enhanced GNN baselines.

**Strengths:**

1) The paper is clearly written and technically sound. It proposes an effective approach to inject LLM-generated auxiliary information into heterophilous edges, enabling more distinguishable node representations; By fine-tuning a small language model (SLM) on class labels, the generated textual annotations are able to incorporate category-related signals.
2) The introduction of active learning to manage LLM inference budgets and improve computational efficiency is novel. The proposed node embedding difference metric (MVRD) provides a principled criterion for edge selection.
3) The method establishes new state-of-the-art performance on node classification across the evaluated datasets.

**Weaknesses:**

1）It is critically important to comprehensively demonstrate the superiority of the proposed method by comparing it with the closely related work LLM4HeG. The results of LLM4HeG should be reported in the main results table, not the appendix.
2) The model relies on more than five hyperparameters, which complicates the search for optimal configurations.
3) The transferability of the fine-tuned small language model (SLM) remains unclear; it is not demonstrated whether an SLM fine-tuned on one dataset can be applied effectively to others.
4) The data partitioning deviates from standard semi-supervised settings, and the impact of training set size on performance is not analyzed.

**Questions:**

1) How do you decide the hyper-parameters? how sensitive your model is w.r.t those hyper-parameters?
2) Does the clustering have any influence the selection of edges?

---

### Official Review · Reviewer_5xn7 · 2025-11-01

**Soundness:** 2
**Presentation:** 3
**Contribution:** 2
**Rating:** 2
**Confidence:** 4

**Summary:**

This paper proposes LEMP4HG, a framework that integrates language models into message passing for heterophilic graph learning. The method uses LM to generate connection analyses with paired node pairs and fuses them with node embeddings through a gating mechanism. And an active learning strategy based on the heuristic MVRD is used to selectively query edges to reduce computational cost.

**Strengths:**

1. The work addresses a meaningful problem, leveraging language models for heterophilic graph learning.

2. The paper provides experiments on multiple datasets and includes ablation and scalability analyses.

3. The paper is well-written.

**Weaknesses:**

1. The integration of LM outputs into message passing builds on earlier LM–GNN fusion works (e.g., LLM4HeG, TAPE) and appears as an incremental engineering variant rather than a fundamental methodological advance.

2. Although the method is designed for heterophilic graphs, on most heterophilic datasets, the proposed model performs worse than SAGE+T and RevGAT+T.

3. The paper introduces an active learning-based selection strategy but does not include comparisons with representative graph active learning methods.

4. There are factual errors in the dataset classification. For example, the Tolokers dataset, which is heterophilic in nature, is mistakenly assigned to the homophilic category in the experiments.

**Questions:**

1. How does the proposed method perform on large-scale graphs such as ogbn-arxiv, ogbn-mag, or ogbn-products?

2. The proposed method is designed for heterophilic graphs, yet the results show inferior performance compared with RevGAT+T and SAGE+T on most heterophilic datasets. Could the authors explain why the complex LM and heuristic designs do not yield better performance?

---

### Note · Authors · 2025-11-26

I have read and agree with the venue's withdrawal policy on behalf of myself and my co-authors.